# MULTI-OBJECTIVE GFLOWNETS

## ABSTRACT

In many applications of machine learning, like drug discovery and material design, the goal is to generate candidates that simultaneously maximize a set of objectives. As these objectives are often conflicting, there is no single candidate that simultaneously maximizes all objectives, but rather a set of Pareto-optimal candidates where one objective cannot be improved without worsening another. Moreover, in practice, these objectives are often under-specified, making the diversity of candidates a key consideration. The existing multi-objective optimization methods focus predominantly on covering the Pareto front, failing to capture diversity in the space of candidates. Motivated by the success of GFlowNets for generation of diverse candidates in a single objective setting, in this paper we consider Multi-Objective GFlowNets (MOGFNs). MOGFNs consist of a novel *Conditional* GFlowNet which models a family of single-objective sub-problems derived by decomposing the multi-objective optimization problem. Our work is the first to empirically demonstrate conditional GFlowNets. Through a series of experiments on synthetic and benchmark tasks, we empirically demonstrate that MOGFNs outperform existing methods in terms of Hypervolume, R2-distance and candidate diversity. We also demonstrate the effectiveness of MOGFNs over existing methods in active learning settings. Finally, we supplement our empirical results with a careful analysis of each component of MOGFNs.

## 1 INTRODUCTION

Decision making in practical applications often involves reasoning about multiple, often conflicting, objectives (Keeney et al., 1993). For example, in drug discovery, the goal is to generate *novel* drug-like molecules that inhibit a target, are easy to synthesize and can safely be used by humans (Dara et al., 2021). Unfortunately, these objectives often conflict – molecules effective against a target might also have adverse effects on humans – so there is no single molecule which maximizes all the objectives simultaneously. Such problems fall under the umbrella of *Multi-Objective Optimization* (MOO; Ehrgott, 2005; Miettinen, 2012), wherein one is interested in identifying *Pareto-optimal* candidates. The set of Pareto-optimal candidates covers all the best tradeoffs among the objectives, i.e., the Pareto front, where each point on that front corresponds to a different set of weights associated with each of the objectives.

*In-silico* drug discovery and material design are typically driven by proxies trained with finite data, which only approximate the problem's true objectives, and therefore include intrinsic epistemic uncertainty associated with their predictions. In such problems, not only it is important to cover the Pareto front, but also to generate sets of *diverse* candidates at each solution of the front so as to increase the likelihood of success in downstream evaluations (Jain et al., 2022).

Generative Flow Networks (GFlowNets; Bengio et al., 2021a;b) are a recently proposed family of probabilistic models which tackle the problem of diverse candidate generation. Contrary to the reward maximization view of reinforcement learning (RL) and Bayesian optimization (BO), GFlowNets sample candidates with probability proportional to the reward. Sampling candidates, as opposed to greedily generating them, implicitly encourages diversity in the generated candidates. GFlowNets have shown promising results in single objective problems of molecule generation (Bengio et al., 2021a) and biological sequence design (Jain et al., 2022).

In this paper, we study *Multi-Objective GFlowNets* (MOGFNs), extensions of GFlowNets which tackle the multi-objective optimization problem. We consider two variants of MOGFNs

– (a) *Preference-Conditional GFlowNets* (MOGFN-PC) which combine *Reward-Conditional GFlowNets* (Bengio et al., 2021b) with Weighted Sum Scalarization (Ehrgott, 2005) and (b) MOGFN-AL, an extension of GFlowNet-AL (Jain et al., 2022) for multi-objective active learning settings. We empirically demonstrate the advantage of MOGFNs over existing approaches on a variety of high-dimensional multi-objective optimization tasks: the generation of small molecules, DNA aptamer sequences and fluorescent proteins. Our contributions are as follows:

**C1** We demonstrate how two variants of GFlowNets – MOGFN-PC and MOGFN-AL – can be applied to multi-objective optimization. Our work is the first successful empirical validation of Reward-Conditional GFlowNets (Bengio et al., 2021b).

**C2** Through a series of experiments on molecule generation and sequence generation we demonstrate that MOGFN-PC generates *diverse* Pareto-optimal candidates.

**C3** In a challenging active learning task for designing fluorescent proteins, we show that MOGFN-AL results in significant improvements to sample-efficiency and diversity of generated candidates.

**C4** We perform a thorough analysis of the main components of MOGFNs to provide insights into design choices that affect performance.

## 2 BACKGROUND

### 2.1 MULTI-OBJECTIVE OPTIMIZATION

Multi-objective optimization (MOO) involves finding a set of feasible candidates $x^\star \in \mathcal{X}$ which all simultaneously maximize a set of objectives:

$$\max_{x \in \mathcal{X}} \left( R_1(x), \ldots, R_d(x) \right). \tag{1}$$

In general, the objectives being optimized can be conflicting such that there is no single $x^\star$ which simultaneously maximizes all objectives. Consequently, the concept of *Pareto optimality* is adopted in MOO, giving rise to a *set of solutions* trading off the objectives in different ways.

Given $x_1, x_2 \in \mathcal{X}$, $x_1$ is said to *dominate* $x_2$, written $(x_1 \succ x_2)$, iff $R_i(x_1) \geq R_i(x_2) \ \forall i \in \{1, \ldots, d\}$ and $\exists k \in \{1, \ldots, d\}$ such that $R_k(x_1) > R_k(x_2)$. A candidate $x^\star$ is *Pareto-optimal* if there exists no other solution $x' \in \mathcal{X}$ which dominates $x^\star$. In other words, for a Pareto-optimal candidate it is impossible to improve one objective without sacrificing another. The *Pareto set* is the set of all Pareto-optimal candidates in $\mathcal{X}$, and the *Pareto front* is defined as the image of the Pareto set in objective-space. It is important to note that since the objectives being optimized in general might not be *injective*, any point on the Pareto front can be the image of several candidates in the Pareto set. This introduces a notion of *diversity in the candidate space*, capturing all the candidates corresponding to a point on the Pareto front, that is critical for applications such as drug discovery.

While there are several paradigms for tackling the MOO problem (Ehrgott, 2005; Miettinen, 2012; Pardalos et al., 2017), we consider *Scalarization*, where the multi-objective problem is decomposed into simpler single-objective problems, as it is well suited for the GFlowNet formulation introduced in Section 3.1. A set of weights (preferences) $\omega_i$ are assigned to the objectives $R_i$, such that $\omega_i \geq 0$ and $\sum_{i=1}^{d} \omega_i = 1$. The MOO problem in Equation 1 is then decomposed into solving single-objective sub-problems of the form $\max_{x \in \mathcal{X}} R(x|\omega)$, where $R$ is a scalarization function.

*Weighted Sum Scalarization*, $R(x|\omega) = \sum_{i=1}^{d} \omega_i R_i(x)$ is a widely used scalarization function which results in Pareto optimal candidates for problems with a convex Pareto front (Ehrgott, 2005). *Weighted Tchebycheff*, $R(x|\omega) = \min_{1 \leq i \leq d} \omega_i |R_i(x) - z_i^\star|$, where $z_i^\star$ denotes some ideal value for objective $R_i$, results in Pareto optimal solutions even for problems with a non-convex Pareto front (Pardalos et al., 2017). See Appendix B for more discussion on scalarization. In summary, using scalarization, the MOO problem can be viewed as solving a family of single-objective optimization problems.

### 2.2 GFLOWNETS

Generative Flow Networks (Bengio et al., 2021a;b) are a family of probabilistic models which generate, through a sequence of steps, *compositional* objects $x \in \mathcal{X}$ with probability proportional to a given reward $R : \mathcal{X} \to \mathbb{R}^+$. The sequential construction of $x \in \mathcal{X}$ can be described as a trajectory

$\tau \in \mathcal{T}$ in a weighted directed acyclic graph (DAG)[1] $\mathcal{G} = (\mathcal{S}, \mathcal{E})$, starting from an empty object $s_0$ and following actions $a \in \mathcal{A}$ as building blocks. The nodes $\mathcal{S}$ of this graph (states) correspond to the set of all possible objects that can be constructed using sequences of actions in $\mathcal{A}$. An edge $s \xrightarrow{a} s' \in \mathcal{E}$ indicates that action $a$ at state $s$ leads to state $s'$.

The forward policy $P_F(-|s)$ is a distribution over the children of state $s$. $x$ can be generated by starting at $s_0$ and sampling a sequence of actions iteratively from $P_F$. Similarly, the backward policy $P_B(-|s)$ is a distribution over the parents of state $s$ and can generate backward trajectories starting at any state $x$, e.g., iteratively sampling from $P_B$ starting at $x$ shows a way $x$ could have been constructed. Let $\pi(x)$ be the marginal likelihood of sampling trajectories terminating in $x$ following $P_F$, and partition function $Z = \sum_{x \in \mathcal{X}} R(x)$. The learning problem solved by GFlowNets is to estimate $P_F$ such that $\pi(x) \propto R(x)$. This is achieved using learning objectives like *trajectory balance* (TB; Malkin et al., 2022), to learn $P_F(-|s;\theta), P_B(-|s;\theta), Z_\theta$ which approximate the forward and backward policies and partition function, parameterized by $\theta$. We refer the reader to Bengio et al. (2021b); Malkin et al. (2022) for a more thorough introduction to GFlowNets.

## 3 MULTI-OBJECTIVE GFLOWNETS

We broadly categorize *Multi-Objective GFlowNets* (MOGFNs) as GFlowNets which solve a family of sub-problems derived from a *Multi-Objective Optimization* (MOO) problem. We first consider solving a family of MOO sub-problems simultaneously with preference-conditional GFlowNets, followed by MOGFN-AL, which solves a sequence of MOO sub-problems.

### 3.1 PREFERENCE-CONDITIONAL GFLOWNETS

Whereas a GFlowNet learns how to sample according to a single reward function, *reward-conditional GFlowNets* (Bengio et al., 2021b) are a generalization of GFlowNets that *simultaneously* model a family of distributions associated with a corresponding family of reward functions. Let $\mathcal{C}$ denote a set of values $c$, with each $c \in \mathcal{C}$ inducing a unique reward function $R(x|c)$. We can define a *family* of weighted DAGs $\{\mathcal{G}_c = (\mathcal{S}_c, \mathcal{E}), \ c \in \mathcal{C}\}$ which describe the construction of $x \in \mathcal{X}$, with conditioning information $c$ available at all states in $\mathcal{S}_c$.

We denote $P_F(-|s, c)$ and $P_B(-|s', c)$ as the *conditional* forward and backward policies, $Z(c) = \sum_{x \in \mathcal{X}} R(x|c)$ as the *conditional* partition function and $\pi(x|c)$ as the marginal likelihood of sampling trajectories $\tau$ from $P_F$ terminating in $x$ given $c$. The learning objective in reward-conditional GFlowNets is thus estimating $P_F(-|s, c)$ such that $\pi(x|c) \propto R(x|c)$. We refer the reader to Bengio et al. (2021b) for a more formal discussion on conditional GFlowNets.

Recall from Section 2.1 that MOO problems can be decomposed into a family of single-objective problems each defined by a preference $\omega$ over the objectives. Thus, we can employ reward-conditional GFlowNets to model the family of reward functions by using as the conditioning set $\mathcal{C}$ the $d$-simplex $\Delta^d$ spanned by the preferences $\omega$ over $d$ objectives.

*Preference-conditional GFlowNets* (MOGFN-PC) are reward-conditional GFlowNets conditioned on the preferences $\omega \in \Delta^d$ over a set of objectives $\{R_1(x), \ldots, R_d(x)\}$. In other words, MOGFN-PC model the family of reward functions $R(x|\omega)$ where $R(x|\omega)$ itself corresponds to a *scalarization* of the MOO problem. We consider three scalarization techniques, which are discussed in Appendix B:

- Weighted-sum (WS) (Ehrgott, 2005): $R(x|\omega) = \sum_{i=1}^{d} \omega_i R_i(x)$
- Weighted-log-sum (WL): $R(x|\omega) = \prod_{i=1}^{d} R_i(x)^{\omega_i}$
- Weighted-Tchebycheff (WT) (Choo & Atkins, 1983): $R(x|\omega) = \min_{1 \leq i \leq d} \omega_i |R_i(x) - z_i^\star|,$.

MOGFN-PC is not constrained to any scalarization function, and can incorporate any user-defined scalarization scheme that fits the desired optimization needs.

**Training MOGFN-PC** The procedure to train MOGFN-PC, or any reward-conditional GFlowNet, closely follows that of a standard GFlowNet and is described in Algorithm 1. The objective is to learn

---

[1]If the object is constructed in a canonical order (say a string constructed from left to right), $\mathcal{G}$ is a tree.

the parameters $\theta$ of the forward and backward conditional policies $P_F(-|s, \omega; \theta)$ and $P_B(-|s', \omega; \theta)$, and the log-partition function $\log Z_\theta(\omega)$. To this end, we consider an extension of the trajectory balance objective for reward-conditional GFlowNets:

$$\mathcal{L}(\tau, \omega; \theta) = \left( \log \frac{Z_\theta(\omega) \prod_{s \to s' \in \tau} P_F(s'|s, \omega; \theta)}{R(x|\omega) \prod_{s \to s' \in \tau} P_B(s|s', \omega; \theta)} \right)^2. \tag{2}$$

One important component is the distribution $p(\omega)$ used to sample preferences during training. $p(\omega)$ influences the regions of the Pareto front that are captured by MOGFN-PC. In our experiments, we use a Dirichlet($\alpha$) to sample preferences $\omega$ which are encoded with thermometer encoding (Buckman et al., 2018) when input to the policy. Following prior work, we also use an exponent $\beta$ for the reward $R(x|\omega)$, i.e. $\pi(x|\omega) \propto R(x|\omega)^\beta$. This incentivizes the policy to focus on the modes of $R(x|\omega)$, which is critical for generation of high reward and diverse candidates.

**MOGFN-PC and MOReinforce**  MOGFN-PC is closely related to MOReinforce (Lin et al., 2021) in that both learn a preference-conditional policy to sample Pareto-optimal candidates. The key difference is the learning objective: MOReinforce uses a multi-objective version of REINFORCE (Williams, 1992), whereas MOGFN-PC uses a preference-conditional GFlowNet objective as in Equation (2). As discussed in Section 2.1, each point on the *Pareto front* (corresponding to a unique $\omega$) can be the image of multiple candidates in the *Pareto set*. MOReinforce, given a preference $\omega$ will converge to sampling a single candidate that maximizes $R(x|\omega)$. MOGFN-PC, on the other hand, samples from $R(x|\omega)$, which enables generation of diverse candidates from the Pareto set for a given $\omega$. This is a key feature of MOGFN-PC whose advantage we empirically demonstrate in Section 5.

### 3.2   MULTI-OBJECTIVE ACTIVE LEARNING WITH GFLOWNETS

In many practical scenarios, the objective functions of interest are computationally expensive. For instance, in the drug discovery scenario, evaluating objectives such as the binding energy to a target even in simulations can take several hours. Sample-efficiency, in terms of number of evaluations of the objective functions, and diversity of candidates, thus become critical in such scenarios. Black-box optimization approaches involving active learning (Zuluaga et al., 2013), particularly multi-objective Bayesian optimization (MOBO) methods (Shah & Ghahramani, 2016; Garnett, 2022) are powerful approaches in these settings.

MOBO uses a probabilistic model to approximate the objectives $\mathbf{R} = \{R_1 \ldots R_d\}$ and leverages the epistemic uncertainty in the predictions of the model as a signal for prioritizing potentially useful candidates. The optimization is performed over $M$ rounds, where each round $i$ consists of generating a batch of candidates $\mathcal{B}$ given all the candidates $\mathcal{D}_i$ proposed in the previous rounds. The batch $\mathcal{B}$ is then evaluated using the true objective functions. The candidates are generated in each round by maximizing an *acquisition function* $a$ which combines the predictions with their epistemic uncertainty into a single scalar utility score. We note that each round is effectively a scalarization of the MOO problem, and as such it may be decomposed into each round's single objective problem.

We broadly define *MOGFN-AL* as approaches which use GFlowNets to generate candidates in each round of an active learning loop for multi-objective optimization. MOGFN-AL tackles MOO through a sequence of single-objective sub-problems defined by acquisition function $a$. As such, MOGFN-AL can be viewed as a multi-objective extension of GFlowNet-AL (Jain et al., 2022). In this work, we consider an instantiation of MOGFN-AL for biological sequence design summarized in Algorithm 2 (Appendix A), building upon the framework proposed by Stanton et al. (2022).

We start with an initial dataset $\mathcal{D}_0 = (x_i, y_i)_{i=1}^N$ of candidates $x_i \in \mathcal{X}$ and their evaluation with the true objectives $y_i = \mathbf{R}(x)$. $\mathcal{D}_i$ is used to train a surrogate probabilistic model (proxy) of the true objectives $\hat{f} : \mathcal{X} \to \mathbb{R}^d$, which we parameterize as a multi-task Gaussian process (Shah & Ghahramani, 2016) with a deep kernel (DKL GP; Maddox et al., 2021a;b). Using this proxy, the acquisition function defines the utility to be maximized $a : \mathcal{X} \times \mathcal{F} \to \mathbb{R}$, where $\mathcal{F}$ denotes the space of functions represented by DKL GPs. In our work we use as acquisition function $a$ *noisy expected hypervolume improvement* (NEHVI; Daulton et al., 2020).

We use GFlowNets to propose candidates at each round $i$ by generating mutations for candidates $x \in \hat{\mathcal{P}}_i$ where $\hat{\mathcal{P}}_i$ is the set of non-dominated candidates in $\mathcal{D}_i$. Given a sequence $x$, the GFlowNet

generates a set of mutations $m = \{(l_i, v_i)\}_{i=1}^T$ where $l \in \{1, \ldots, |x|\}$ is the location to be replaced and $v \in \mathcal{A}$ is the token to replace $x[l]$ while $T$ is the number of mutations. This set is generated sequentially such that each mutation is sampled from $P_F$ conditioned on $x$ and the mutations sampled so far $\{(l_i, v_i)\}$. Let $x'_m$ be the sequence resulting from mutations $m$ on sequence $x$. The reward for a set of sampled mutations for $x$ is the value of the acquisition function on $x'_m$, $R(m, x) = a(x'_m|\hat{f})$. This approach of generating mutations to existing sequences provides an key advantage over generating sequences token-by-token as done in prior work (Jain et al., 2022) – better scaling for longer sequences. We show empirically in Section 5.3 that generating mutations with GFlowNets results in more diverse candidates and faster improvements to the Pareto front than LaMBO (Stanton et al., 2022).

## 4  RELATED WORK

**Evolutionary Algorithms (EA)**   Traditionally, evolutionary algorithms such as NSGA-II have been widely used in various multi-objective optimization problems (Ehrgott, 2005; Konak et al., 2006; Blank & Deb, 2020). More recently, Miret et al. (2022) incorporated graph neural networks into evolutionary algorithms enabling them to tackle large combinatorial spaces. Unlike MOGFNs, evolutionary algorithms do not leverage any type of data, including past experiences, and therefore are required to solve each instance of a MOO from scratch rather than by amortizing computation during training in order to quickly generate solutions at run-time. Evolutionary algorithms, however, can be augmented with MOGFNs for generating mutations to improve efficiency, as in Section 3.2.

**Multi-Objective Reinforcement Learning**   MOO problems have also received significant interest in the reinforcement learning (RL) literature (Hayes et al., 2022). Traditional approaches broadly consist of learning sets of Pareto-dominant policies (Roijers et al., 2013; Van Moffaert & Nowé, 2014; Reymond et al., 2022). Recent work has focused on extending Deep RL algorithms for multi-objective settings such as Envelope-MOQ (Yang et al., 2019), MO-MPO (Abdolmaleki et al., 2020; 2021) , and MOReinforce (Lin et al., 2021). A general shortcoming of RL based approaches is that they only discover a single mode of the reward function, and thus cannot generate diverse candidates, which also persists in the multi-objective setting. In contrast, MOGFNs sample candidates proportional to the reward, implicitly resulting in diverse candidates.

**Multi-Objective Bayesian Optimization (MOBO)**   Bayesian optimization (BO) has been used in the context of MOO when the objectives are expensive to evaluate and sample-efficiency is a key consideration. MOBO approaches consist of learning a surrogate model of the true objective functions, which is used to define an acquisition function such as expected hypervolume improvement (Emmerich et al., 2011; Daulton et al., 2020; 2021) and max-value entropy search (Belakaria et al., 2019), as well as scalarization-based approaches (Paria et al., 2020; Zhang & Golovin, 2020). Stanton et al. (2022) proposed LaMBO, which uses language models in conjunction with BO for multi-objective sequence design problems. The key drawbacks of MOBO approaches are that they do not consider the need for diversity in generated candidates and that they mainly consider continuous state spaces. As we discuss in Section 3.2, MOBO approaches can be augmented with GFlowNets for *diverse* candidate generation in discrete spaces.

**Other Works**   Zhao et al. (2022) introduced LaMOO which tackles the MOO problem by iteratively splitting the candidate space into smaller regions, whereas Daulton et al. (2022) introduce MORBO, which performs BO in parallel on multiple local regions of the candidate space. Both these methods, however, are limited to continuous candidate spaces.

## 5  EMPIRICAL RESULTS

In this section, we present our empirical findings across a wide range of tasks ranging from sequence design to molecule generation.The experiments cover two distinct classes of problems in the context of GFlowNets: where $\mathcal{G}$ is a DAG and where it is a tree. Through our experiments, we aim to answer the following questions:

**Q1** *Can MOGFNs model the preference-conditional reward distribution?*

**Q2** *Can MOGFNs sample Pareto-optimal candidates?*

**Q3** *Are candidates sampled by MOGFNs* diverse*?*

**Q4** *Do MOGFNs scale to high-dimensional problems relevant in practice?*

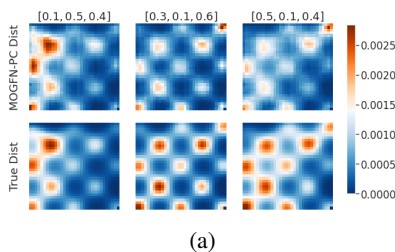 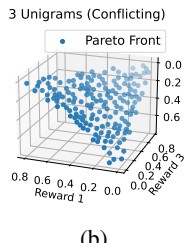 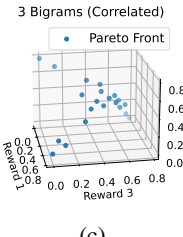

|   (a)   |   (b)   |   (c)   |

Figure 1: *(a)* The distribution learned by MOGFN-PC (Top) almost exactly matches the ground truth distribution (Bottom), in particular capturing all the modes, on hypergrid of size $32 \times 32$ with 3 objectives. *(b)* and *(c)* illustrate the Pareto front of candidates generated by MOGFN-PC with conflicting and correlated objectives respectively.

**Metrics:** We rely on standard metrics such as the **Hypervolume** (HV) and $R_2$ **indicators**, as well as the **Generational Distance+** (GD+). To measure diversity we use the **Top-K Diversity** and **Top-K Reward** metrics of Bengio et al. (2021a). We detail all metrics in Appendix D. For all our empirical evaluations we follow the same protocol. First, we sample a set of preferences which are fixed for all the methods. For each preference we sample 128 candidates from which we pick the top 10, compute their scalarized reward and diversity, and report the averages over preferences. We then use these samples to compute the HV and $R_2$ indicators. We pick the best hyperparameters for all methods based on the HV and report the mean and standard deviation over 3 seeds for all quantities.

**Baselines**: We consider the closely related MOReinforce (Lin et al., 2021) as a baseline. We also study its variants MOSoftQL and MOA2C which use Soft Q-Learning (Haarnoja et al., 2017) and A2C (Mnih et al., 2016) in place of REINFORCE. We also compare against Envelope-MOQ (Yang et al., 2019), another popular multi-objective reinforcement learning method. For fragment-based molecule generation we consider an additional baseline MARS (Xie et al., 2021), a relevant MCMC approach for this task. To keep comparisons fair, we omit baselines like LaMOO (Zhao et al., 2022) and MORBO (Daulton et al., 2022) as they are designed for continuous spaces and rely on latent representations from pre-trained models for discrete tasks like molecule generation.

## 5.1 SYNTHETIC TASKS

### 5.1.1 HYPER-GRID

We first study the ability of MOGFN-PC to capture the preference-conditional reward distribution in a multi-objective version of the HyperGrid task from Bengio et al. (2021a). The goal here is to navigate proportional to a reward within a HyperGrid. We consider the following objectives for our experiments: `brannin(x)`, `currin(x)`, `shubert(x)`[2].

Since the state space is small, we can compute the distribution learned by MOGFN-PC in closed form. In Figure 1a, we visualize $\pi(x|\omega)$, the distribution learned by MOGFN-PC conditioned on a set of fixed preference vectors $\omega$ and contrast it with the true distribution $R(x|\omega)$ in a $32 \times 32$ hypergrid with 3 objectives. We observe that $\pi(-|\omega)$ and $R(-|\omega)$ are very similar. To quantify this, we compute $\mathbb{E}_x \left[ |\pi(x|\omega) - R(x|\omega)/Z(\omega)| \right]$ averaged over a set of 64 preferences, and find a difference of about $10^{-4}$. Note that MOGFN-PC is able to capture all the modes in the distribution, which suggests the candidates sampled from $\pi$ would be diverse. Further, we compute the GD+ metric for the Pareto front of candidates generated with MOGFN-PC, which comes up to an average value of 0.42. For more details about the task and the additional results, refer to Appendix E.1.

### 5.1.2 N-GRAMS TASK

We consider version of the synthetic sequence design task from Stanton et al. (2022). The task consists of generating strings with the objectives given by occurrences of a set of $d$ n-grams.

In the results summarized in Table 1, we consider 3 Bigrams (with common characters in the bigrams resulting in correlated objectives) and 3 Unigrams (conflicting objectives) as the objectives. MOGFN-PC outperforms the baselines in terms of the MOO objectives while generating diverse candidates.

---

[2]We present additional results with more objectives in Appendix E.1

Table 1: **N-Grams Task:** Diversity and Pareto performance of various algorithms on for the 3 Bigrams and 3 Unigrams tasks with MOGFN-PC achieving superior Pareto performance.

| Algorithm | 3 Bigrams | | | | 3 Monograms | | | |
|---|---|---|---|---|---|---|---|---|
| | Reward ($\uparrow$) | Diversity ($\uparrow$) | HV ($\uparrow$) | $R_2$ ($\downarrow$) | Reward ($\uparrow$) | Diversity ($\uparrow$) | HV ($\uparrow$) | $R_2$ ($\downarrow$) |
| Envelope-MOQ | $0.05_{\pm 0.04}$ | $0_{\pm 0}$ | $0.012_{\pm 0.013}$ | $19.66_{\pm 0.66}$ | $0.08_{\pm 0.015}$ | $0_{\pm 0}$ | $0.023_{\pm 0.011}$ | $21.18_{\pm 0.72}$ |
| MOReinforce | $0.12_{\pm 0.02}$ | $0_{\pm 0}$ | $0.015_{\pm 0.021}$ | $20.32_{\pm 0.93}$ | $0.03_{\pm 0.001}$ | $0_{\pm 0}$ | $0.036_{\pm 0.009}$ | $21.04_{\pm 0.51}$ |
| MOSoftQL | $0.28_{\pm 0.03}$ | $21.09_{\pm 0.65}$ | $0.093_{\pm 0.025}$ | $15.79_{\pm 0.23}$ | $0.36_{\pm 0.01}$ | $23.131_{\pm 0.6736}$ | $0.105_{\pm 0.014}$ | $12.80_{\pm 0.26}$ |
| MOGFN-PC | $0.44_{\pm 0.01}$ | $19.79_{\pm 0.08}$ | $0.220_{\pm 0.017}$ | $9.97_{\pm 0.45}$ | $0.38_{\pm 0.00}$ | $22.71_{\pm 0.24}$ | $0.121_{\pm 0.015}$ | $11.39_{\pm 0.17}$ |

Since the objective counts occurrences of n-grams, the diversity is limited by the performance, i.e. high scoring sequences will have lower diversity, explaining higher diversity of MOSoftQL. We note that the MOReinforce and Envelope-MOQ baselines struggle in this task potentially due to longer trajectories with sparse rewards. MOGFN-PC adequately models the trade-off between conflicting objectives in the 3 Monograms task as illustrated by the Pareto front of generated candidates in Figure 1b. For the 3 Bigrams task with correlated objectives, Figure 1c demonstrates MOGFN-PC generates candidates which can simultaneously maximize multiple objectives. We refer the reader to Appendix E.2 for more task details and additional results with different number of objectives and varying sequence length.

## 5.2 BENCHMARK TASKS

### 5.2.1 QM9

We first consider a small-molecule generation task based on the QM9 dataset (Ramakrishnan et al., 2014). We generate molecules atom-by-atom and bond-by-bond with up to 9 atoms and use 4 reward signals. The main reward is obtained via a MXMNet (Zhang et al., 2020) proxy trained on QM9 to predict the HOMO-LUMO gap. The other rewards are Synthetic Accessibility (SA), a molecular weight target, and a molecular logP target. Rewards are normalized to between 0 and 1, but the gap proxy can exceed 1, and so is clipped at 2. We train the models with 1M molecules and present the results in Table 2, showing that MOGFN-PC outperforms all baselines in terms of Pareto performance and diverse candidate generation.

Table 2: **Atom-based QM9 task:** MOGFN-PC exceeds Diversity and Pareto performance on QM9 task with HUMO-LUMO gap, SA, QED and molecular weight objectives compared to baselines.

| Algorithm | Reward ($\uparrow$) | Diversity ($\uparrow$) | HV ($\uparrow$) | $R_2$ ($\downarrow$) |
|---|---|---|---|---|
| MOA2C (Mnih et al., 2016) | $0.61_{\pm 0.05}$ | $0.39_{\pm 0.28}$ | $1.16_{\pm 0.08}$ | $6.28_{\pm 0.67}$ |
| Envelope QL (Yang et al., 2019) | $0.65_{\pm 0.06}$ | $0.85_{\pm 0.01}$ | $1.26_{\pm 0.05}$ | $5.80_{\pm 0.20}$ |
| MOReinforce (Lin et al., 2021) | $0.57_{\pm 0.12}$ | $0.53_{\pm 0.08}$ | $1.35_{\pm 0.01}$ | $4.65_{\pm 0.03}$ |
| MOGFN-PC | $0.76_{\pm 0.00}$ | $0.93_{\pm 0.00}$ | $1.40_{\pm 0.18}$ | $2.44_{\pm 1.88}$ |

### 5.2.2 FRAGMENT-BASED MOLECULE GENERATION

We evaluate our method on the fragment-based (Kumar et al., 2012) molecular generation task of Bengio et al. (2021a), where the task is to generate molecules by linking fragments to form a junction tree (Jin et al., 2020). The main reward function is obtained via a pretrained proxy, available from Bengio et al. (2021a), trained on molecules docked with AutodockVina (Trott & Olson, 2010) for the sEH target. The other rewards are based on Synthetic Accessibility (SA), drug likeness (QED), and a molecular weight target. We detail the reward construction in Appendix E.4. Similarly to QM9, we train MOGFN-PC to generate 1M molecules and report the results in Table 3. We observe that MOGFN-PC is consistently outperforming baselines not only in terms of HV and $R_2$, but also candidate diversity score. Note that we do not report reward and diversity scores for MARS, since the lack of preference conditioning would make it an unfair comparison.

### 5.2.3 DNA SEQUENCE GENERATION

As a practical domain where the GFlowNet graph is a tree, we consider the generation of DNA aptamers, single-stranded nucleotide sequences that are popular in biological polymer design due to their specificity and affinity as sensors in crowded biochemical environments (Zhou et al., 2017; Corey et al., 2022; Yesselman et al., 2019; Kilgour et al., 2021). We generate sequences by adding one nucleobase (A, C, T or G) at a time, with a maximum length of 60 bases. We consider three objectives:

Table 3: **Fragment-based Molecule Generation Task:** Diversity and Pareto performance on the Fragment-based drug design task with sEH, QED, SA and molecular weight objectives.

| Algorithm | Reward ($\uparrow$) | Diversity ($\uparrow$) | HV ($\uparrow$) | $R_2$ ($\downarrow$) |
|---|---|---|---|---|
| MOReinforce (Lin et al., 2021) | $0.41_{\pm 0.07}$ | $0.01_{\pm 0.007}$ | $0_{\pm 0}$ | $9.88_{\pm 1.06}$ |
| MARS (Xie et al., 2021) | – | – | $0.85_{\pm 0.008}$ | $1.94_{\pm 0.03}$ |
| MOA2C (Mnih et al., 2016) | $0.76_{\pm 0.16}$ | $0.48_{\pm 0.39}$ | $0.75_{\pm 0.01}$ | $3.35_{\pm 0.02}$ |
| Envelope QL (Yang et al., 2019) | $0.70_{\pm 0.10}$ | $0.15_{\pm 0.05}$ | $0.74_{\pm 0.01}$ | $3.51_{\pm 0.10}$ |
| MOGFN-PC | $0.89_{\pm 0.05}$ | $0.75_{\pm 0.01}$ | $0.90_{\pm 0.01}$ | $1.86_{\pm 0.08}$ |

the free energy of the secondary structured calculated with the software NUPACK (Zadeh et al., 2011), the number of base pairs and the inverse of the sequence length to favour shorter sequences.

We report the results in Table 4. In this case, the best Pareto performance is obtained by the multi-objective RL algorithm MOReinforce (Lin et al., 2021). However, it achieves so by finding a quasi-trivial solution with the pattern GCGCGC... for most lengths, yielding very low diversity. In contrast, MOGFN-PC obtains much higher diversity and Top-K rewards but worse Pareto performance. An extended discussion, ablation study and further details are provided in Appendix E.5.

Table 4: **DNA Sequence Design Task:** Diversity and Pareto performance of various algorithms on DNA sequence generation task with free energy, number of base pairs and inverse sequence length objectives.

| Algorithm | Reward ($\uparrow$) | Diversity ($\uparrow$) | HV ($\uparrow$) | $R_2$ ($\downarrow$) |
|---|---|---|---|---|
| Envelope-MOQ (Yang et al., 2019) | $0.238_{\pm 0.042}$ | $0.0_{\pm 0.0}$ | $0.163_{\pm 0.013}$ | $5.657_{\pm 0.673}$ |
| MOReinforce (Lin et al., 2021) | $0.105_{\pm 0.002}$ | $0.6178_{\pm 0.209}$ | $0.629_{\pm 0.002}$ | $1.925_{\pm 0.003}$ |
| MOSoftQL | $0.446_{\pm 0.010}$ | $32.130_{\pm 0.542}$ | $0.163_{\pm 0.014}$ | $5.565_{\pm 0.170}$ |
| MOGFN-PC | $0.682_{\pm 0.021}$ | $18.131_{\pm 0.981}$ | $0.517_{\pm 0.006}$ | $2.432_{\pm 0.002}$ |

## 5.3 ACTIVE LEARNING

Finally, to evaluate MOGFN-AL, we consider the Proxy RFP task from Stanton et al. (2022), with the aim of discovering novel proteins with red fluorescence properties, optimizing for folding stability and solvent-accessible surface area. We adopt all the experimental details (described in Appendix E.6) from Stanton et al. (2022), using MOGFN-AL for candidate generation. In addition to LaMBO, we use a model-free (NSGA-2) and model-based EA from Stanton et al. (2022) as baselines. We observe in Figure 2a that MOGFN-AL results in significant gains to the improvement in Hypervolume relative to the initial dataset, in a given budget of black-box evaluations. In fact, MOGFN-AL is able to match the performance of LaMBO within about half the number of black-box evaluations.

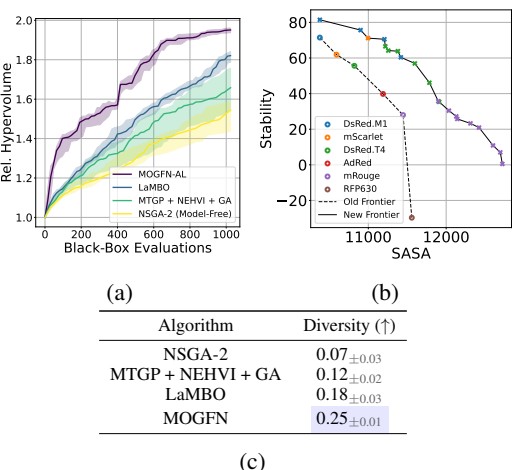

(a)        (b)

| Algorithm | Diversity ($\uparrow$) |
|---|---|
| NSGA-2 | $0.07_{\pm 0.03}$ |
| MTGP + NEHVI + GA | $0.12_{\pm 0.02}$ |
| LaMBO | $0.18_{\pm 0.03}$ |
| MOGFN | $0.25_{\pm 0.01}$ |

(c)

Figure 2: (a) MOGFN-AL demonstrates substantial advantage in terms of Relative Hypervolume and (b) Pareto frontier of candidates generated by MOGFN-AL dominates the Pareto front of the initial dataset. (c) MOGFN-AL is particularly strong in terms of diversity of candidates.

Figure 2b illustrates that the Pareto frontier of candidates generated with MOGFN-AL, which dominates the Pareto frontier of the initial dataset. As we the candidates are generated by mutating sequences in the existing Pareto front, we also highlight the sequences that are mutations of each seqeunce in the initial dataset with the same color. To quantify the diversity of the generated candidates we measure the average e-value from DIAMOND (Buchfink et al., 2021) between the initial Pareto front and the Pareto frontier of generated candidates. Table 2c shows that MOGFN-AL generates candidates that are more diverse than the baselines.

# 6 ANALYSIS

In this section, we isolate the important components of MOGFN-PC: the distribution $p(\omega)$ for sampling preferences during training, the reward exponent $\beta$ and the reward scalarization $R(x|\omega)$ to understand the impact of each component on Pareto performance and diversity. We consider the 3 Bigrams task discussed in Section 5.1.2 and the fragment-based molecule generation task from Section 5.2.1 for this analysis and provide further results in the Appendix.

**Impact of** $p(\omega)$  To examine the effect of $p(\omega)$, which controls the coverage of the Pareto front, we set it to Dirichlet($\alpha$) and vary $\alpha \in \{0.1, 1, 10\}$. This results in $\omega$ being sampled from different regions of $\Delta^d$. Specifically, $\alpha = 1$ corresponds to a uniform distribution over $\Delta^d$, $\alpha > 1$ is skewed towards the center of $\Delta^d$ whereas $\alpha < 1$ is skewed towards the corners of $\Delta^d$. In Table 5 and Table 6 we observe that $\alpha = 1$ results in the best performance. Despite the skewed distribution with $\alpha = 0.1$ and $\alpha = 10$, we still achieve performance close to that of $\alpha = 1$ indicating that MOGFN-PC is able to interpolate to preferences not sampled during training. Note that diversity is not affected significantly by $p(\omega)$.

Table 5: **N-grams:** Analysing the impact of $\alpha$, $\beta$ and $R(x|\omega)$ on the performace of MOGFN-PC

| Metrics | Effect of $p(\omega)$ | | | Effect of $\beta$ | | | Choice of $R(x \mid \omega)$ | | |
|---|---|---|---|---|---|---|---|---|---|
| | Dir(0.1) | Dir(1) | Dir(10) | 16 | 32 | 48 | WS | WL | WT |
| Reward ($\uparrow$) | $0.38_{\pm 0.02}$ | $0.44_{\pm 0.01}$ | $0.37_{\pm 0.03}$ | $0.22_{\pm 0.004}$ | $0.36_{\pm 0.008}$ | $0.44_{\pm 0.01}$ | – | – | – |
| Diversity ($\uparrow$) | $18.82_{\pm 0.41}$ | $19.79_{\pm 0.08}$ | $18.56_{\pm 0.75}$ | $29.23_{\pm 0.34}$ | $23.86_{\pm 0.35}$ | $19.79_{\pm 0.08}$ | $19.79_{\pm 0.08}$ | $24.87_{\pm 0.62}$ | $22.51_{\pm 0.34}$ |
| Hypervolume ($\uparrow$) | $0.17_{\pm 0.015}$ | $0.22_{\pm 0.017}$ | $0.18_{\pm 0.009}$ | $0.06_{\pm 0.006}$ | $0.14_{\pm 0.008}$ | $0.22_{\pm 0.02}$ | $0.22_{\pm 0.017}$ | $0.051_{\pm 0.010}$ | $0.097_{\pm 0.021}$ |
| $R_2$ ($\downarrow$) | $10.95_{\pm 0.21}$ | $9.97_{\pm 0.45}$ | $10.52_{\pm 0.13}$ | $17.00_{\pm 0.23}$ | $12.46_{\pm 0.26}$ | $9.97_{\pm 0.45}$ | $9.97_{\pm 0.45}$ | $21.54_{\pm 0.71}$ | $18.17_{\pm 0.32}$ |

**Impact of** $\beta$  During training $\beta$, controls the concentration of the reward density around modes of the distribution. For large values of $\beta$ the reward density around the modes become more peaky and vice-versa. In Table 5 and Table 6 we present the results obtained by varying $\beta \in \{16, 32, 48\}$. As $\beta$ increases, MOGFN-PC is incentivized to generate samples closer to the modes of $R(x|\omega)$, resulting in better Pareto performance. However, with high $\beta$ values, the reward density is concentrated close to the modes and there is a negative impact on the diversity of the candidates.

**Choice of scalarization** $R(x|\omega)$  Next, we analyse the effect of the scalarization defining $R(x|\omega)$ used for training. The set of $R(x|\omega)$ for different $\omega$ specifies the family of MOO sub-problems and thus has a critical impact on the Pareto performance. Table 5 and Table 6 include results for the Weighted Sum (WS), Weighted-log-sum (WL) and Weighted Tchebycheff (WT) scalarizations. Note that we do not compare the Top-K Reward as different scalarizations cannot be compared directly. WS scalarization results in the best performance. WL scalarization on the other hand is not formally guaranteed to cover the Pareto front and consequently results in poor Pareto performance. We suspect the poor performance of WT and WL are in part also due to the harder reward landscapes they induce.

# 7 CONCLUSION

In this work, we have empirically demonstrated the generalization of GFlowNets to *conditional* GFlowNets for multi-objective optimization problems (MOGFN) to promote the generation of diverse optimal candidates. We presented two instantiations of MOGFN: MOGFN-PC, which leverages reward-conditional GFlowNets (Bengio et al., 2021b) to model a family of single-objective sub-problems, and MOGFN-AL, which sequentially solves a set of single-objective problems defined by multi-objective acquisition functions. Finally, we empirically demonstrated the efficacy of MOGFNs for generating diverse Pareto-optimal candidates on sequence and graph generation tasks.

As a limitation, we identify that in certain domains, such as DNA sequence generation, MOGFN generates diverse candidates but currently does not match RL algorithms in terms of Pareto performance. The analysis in Section 6 hints that the distribution of sampling preferences $p(\omega)$ affects the Pareto performance. Since for certain practical applications only a specific region of the Pareto front is of interest, future work may explore gradient based techniques to learn preferences for more structured exploration of the preference space. Within the context of MOFGN-AL, an interesting research avenue is the development of preference-conditional acquisition functions.

**Reproducibility Statement** We include the code necessary to replicate experiments with our submission and provide detailed description of experimental setups in the Appendix. All datasets and pretrained models used are publicly available or included in the supplementary materials.

**Ethics Statement** We acknowledge that as with all machine learning algorithms, there is potential for dual use of multi-objective GFlowNets by nefarious agents. This work was motivated by the application of machine learning to accelerate scientific discovery in areas that can benefit humanity. We explicitly discourage the use of multi-objective GFlowNets in applications that may be harmful to others.

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

# A ALGORITHMS

We summarize the algorithms for MOGFN-PC and MOGFN-AL here.

---

**Algorithm 1:** Training preference-conditional GFlowNets

---

**Input**:

$p(\omega)$: Distribution for sampling preferences;

$\beta$: Reward Exponent;

$\delta$: Mixing Coefficient for uniform actions in sampling policy;

$N$: Number of training steps;

**Initialize**:

$(P_F(s'|s, \omega), P_B(s|s', \omega), \log Z(\omega))$: Conditional GFlowNet with parameters $\theta$;

**for** $i = 1$ *to* $N$ **do**

    Sample preference $\omega \sim p(\omega)$;

    Sample trajectory $\tau$ following policy $\hat{\pi} = (1 - \delta)P_F \ + \delta \text{Uniform}$ ;

    Compute reward $R(x|\omega)^\beta$ for generated samples and corresponding loss $\mathcal{L}(\tau, \omega; \theta)$ as in
      Equation 2;

    Update parameters $\theta$ with gradients from the loss, $\nabla_\theta \mathcal{L}(\tau, \omega)$;

**end**

---

---

**Algorithm 2:** Training MOGFN-AL

---

**Input**:

$\mathbf{R} = \{R_1, \dots, R_d\}$: Oracles to evaluate candidates $x$ and return true objectives
$(R_1(x), \dots, R_d(x))$ ;

$D_0 = \{(x_i, y_i)\}$: Initial dataset with $y_i = \mathbf{R}(x_i)$;

$\hat{f}$: Probabilistic surrogate model to model posterior over $\mathbf{R}$ given a dataset $\mathcal{D}$;

$a(x|\hat{f})$: Acquisition function computing a scalar utility for $x$ given $\hat{f}$;

$\pi_\theta$: Learnable GFlowNet policy;

$b$: Size of candidate batch to be generated;

$N$: Number of active learning rounds;

**Initialize**:

$\hat{f}, \pi_\theta$;

**for** $i = 1$ *to* $N$ **do**

    Fit $\hat{f}$ on dataset $D_{i-1}$;

    Extract the set of non-dominated candidates $\hat{\mathcal{P}}_{i-1}$ from $D_{i-1}$;

    Train $\pi_\theta$ with to generate mutations for $x \in \hat{\mathcal{P}}_i$ using $a(-|\hat{f})$ as the reward;

    Generate batch $\mathcal{B} = \{x'_{1,m_i}, \dots, x'_{b,m_b}\}$ by sampling $x'_i$ from $\hat{\mathcal{P}}_{i-1}$ and applying to it
      mutations $m_i$ sampled from $\pi_\theta$;

    Evaluate batch $\mathcal{B}$ with $\mathbf{R}$ to generate $\hat{D}_i = \{(x_1, \mathbf{R}(x_1)), \dots, (x_b, \mathbf{R}(x_b))\}$;

    Update dataset $D_i = \hat{D}_i \cup D_{i-1}$

**end**

**Result**:

Approximate Pareto set $\hat{\mathcal{P}}_N$

---

# B SCALARIZATION

Scalarization is a popular approach for tackling multi-objective optimization problems. MOGFN-PC can build upon any scalarization approach. We consider three choices. Weighted-sum (WS) scalarization has been widely used in literature. WS finds candidates on the convex hull of the Pareto front (Ehrgott, 2005). Under the assumption that the Pareto front is convex, every Pareto optimal solution is a solution to a weighted sum problem and the solution to every weighted sum problem is Pareto optimal. Weigthed Tchebycheff (WT), proposed by Choo & Atkins (1983) is

an alternative designed for non-convex Pareto fronts. Any Pareto optimal solution can be found by solving the weighted Tchebycheff problem with appropriate weights, and the solutions for any weights correspond to a *weakly* Pareto optimal solution of the original problem (Pardalos et al., 2017). Lin et al. (2021) deomstrated through their empirical results that WT can be used with neural network based policies. The third scheme we consider, Weighted-log-sum (WL) has not been considered in prior work. We hypothesized that in some practical scenarios, we might want to ensure that all objectives are optimized, since, for instance, in WS the scalarized reward can be dominated by a single reward. WL, which considers the weigthed sum in $\log$ space can potentially help with this drawback. However, as discussed in Section 6, in practice WL can be hard to optimize, and lead to poor performance.

## C  ADDITIONAL ANALYSIS

Table 6: **Fragment-based molecule generation:** Analysing the impact of $\alpha$, $\beta$ and $R(x|\omega)$ on the performace of MOGFN-PC

| Metrics | Effect of $p(\omega)$ | | | Effect of $\beta$ | | | | Choice of $R(x \mid \omega)$ | | |
|---|---|---|---|---|---|---|---|---|---|---|
| | Dir(0.1) | Dir(1) | Dir(10) | 16 | 32 | 48 | 96 | WS | WL | WT |
| Reward (↑) | $0.57_{\pm0.04}$ | $0.89_{\pm0.05}$ | $0.58_{\pm0.03}$ | $0.44_{\pm0.02}$ | $0.51_{\pm0.008}$ | $0.55_{\pm0.008}$ | $0.89_{\pm0.05}$ | – | – | – |
| Diversity (↑) | $0.79_{\pm0.01}$ | $0.75_{\pm0.01}$ | $0.75_{\pm0.09}$ | $0.86_{\pm0.006}$ | $0.86_{\pm0.001}$ | $0.85_{\pm0.002}$ | $0.75_{\pm0.01}$ | $0.75_{\pm0.01}$ | $0.82_{\pm0.016}$ | $0.10_{\pm0.002}$ |
| Hypervolume (↑) | $0.67_{\pm0.08}$ | $0.90_{\pm0.01}$ | $0.82_{\pm0.12}$ | $0.59_{\pm0.06}$ | $0.55_{\pm0.001}$ | $0.60_{\pm0.06}$ | $0.90_{\pm0.01}$ | $0.90_{\pm0.01}$ | $0.55_{\pm0.017}$ | $0.71_{\pm0.10}$ |
| $R_2$ (↓) | $2.57_{\pm0.43}$ | $1.86_{\pm0.08}$ | $1.93_{\pm1.12}$ | $5.76_{\pm0.30}$ | $4.46_{\pm0.28}$ | $3.64_{\pm0.19}$ | $1.86_{\pm0.08}$ | $1.86_{\pm0.08}$ | $6.92_{\pm0.18}$ | $11.51_{\pm1.79}$ |

**Can MOGFN-PC match Single Objective GFNs?**    To evaluate how well MOGFN-PC models the family of rewards $R(x|\omega)$, we consider a comparison with single objective GFlowNets. More specifically, we first sample a set of 10 preferences $\omega_1, \dots, \omega_{10}$, and train a standard single objective GFlowNet using the weighted sum scalar reward for each preference. We then generate $N = 128$ candidates from each GFlowNet, throughout training, and compute the mean reward for the top 10 candidates for each preference. We average this top 10 reward across $\{\omega_1, \dots, \omega_{10}\}$, and call it $R_{so}$. We then train MOGFN-PC, and apply the sample procedure with the preferences $\{\omega_1, \dots, \omega_{10}\}$, and call the resulting mean of top 10 rewards $R_{mo}$. We plot the value of the ratio $R_{mo}/R_{so}$ in Figure 3. We observe that the ratio stays close to 1, indicating that MOGFN-PC can indeed model the entire family of rewards simultaneously at least as fast as a single objective GFlowNet could.

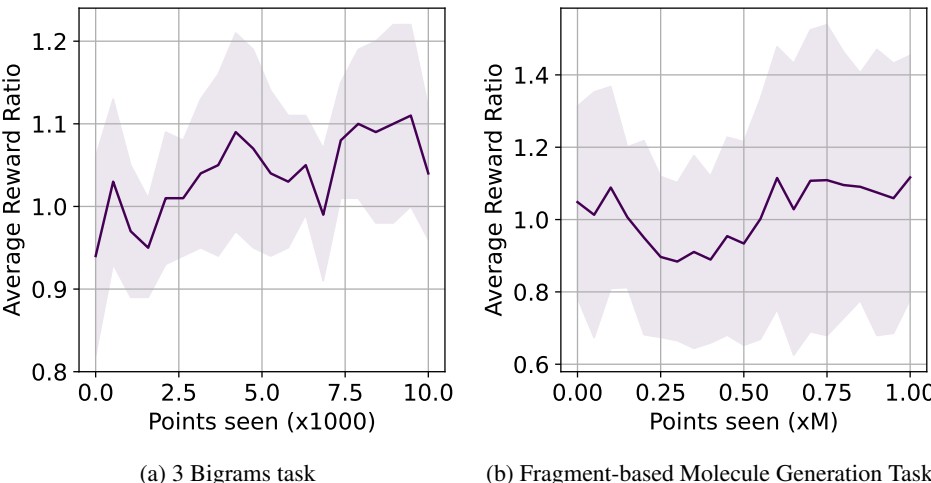

(a) 3 Bigrams task                    (b) Fragment-based Molecule Generation Task

Figure 3: We plot the ratio of rewards $R_{mo}/R_{so}$ for candidates generated with MOGFN-PC ($R_mo$) and single-objective GFlowNets($R_{so}$) for a set of preferences in the (a) 3 Bigrams and (b) Fragment-based molecule generation tasks. We observe that MOGFN-PC matches and occasionally surpasses single objective GFlowNets

**Effect of Model Capacity and Architecture**   Finally we look at the effect of model size in training MOGFN-PC. As MOGFN-PC models a conditional distribution, an entire family of functions as we've described before, we expect capacity to play a crucial role since the amount of information to be learned is higher than for a single-objective GFN. We increase model size in the 3 Bigrams task to see that effect, and see in Table 7 that larger models do help with performance–although the performance plateaus after a point. We suspect that in order to fully utilize the model capacity we might need better training objectives.

Table 7: Analysing the impact of model size on the performance of MOGFN-PC. Each architecture choice for the policy is denoted as A-B-C where A is number of layers, B is the number of hidden units in each layer, and C is the number of attention heads.

| Metrics | Effect of model size | | | |
| --- | --- | --- | --- | --- |
| | 3-64-8 | 3-256-8 | 4-64-8 | 4-256-8 |
| Reward ($\uparrow$) | $0.44_{\pm0.01}$ | $0.47_{\pm0.00}$ | $0.49_{\pm0.03}$ | $0.51_{\pm0.01}$ |
| Diversity ($\uparrow$) | $19.79_{\pm0.08}$ | $17.13_{\pm0.38}$ | $17.53_{\pm0.15}$ | $16.12_{\pm0.04}$ |
| Hypervolume ($\uparrow$) | $0.22_{\pm0.017}$ | $0.255_{\pm0.008}$ | $0.262_{\pm0.003}$ | $0.270_{\pm0.011}$ |
| $R_2$ ($\downarrow$) | $9.97_{\pm0.45}$ | $9.22_{\pm0.25}$ | $8.95_{\pm0.05}$ | $8.91_{\pm0.12}$ |

## D  METRICS

In this section we discuss the various metrics that we used to report the results in Section 5.

1. **Generational Distance Plus (GD +)** (Ishibuchi et al., 2015): This metric measures the euclidean distance between the solutions of the Pareto approximation and the true Pareto front by taking the dominance relation into account. To calculate **GD+** we require the knowledge of the true Pareto front and hence we only report this metric for Hypergrid experiments (Section 5.1.1)

2. **Hypervolume (HV) Indicator** (Fonseca et al., 2006): This is a standard metric reported in MOO works which measures the volume in the objective space with respect to a reference point spanned by a set of non-dominated solutions in Pareto front approximation.

3. **$R_2$ Indicator** (Hansen & Jaszkiewicz, 1994): $R_2$ provides a monotonic metric comparing two Pareto front approximations using a set of uniform reference vectors and a utopian point $z^*$ representing the ideal solution of the MOO.

   This metric provides a monotonic reference to compare different Pareto front approximations relative to a utopian point. Specifically, we define a set of uniform reference vectors $\lambda \in \Lambda$ that cover the space of the MOO and then calculate:$R_2(\Gamma, \Lambda, z^*) = \frac{1}{|\Lambda|} \sum_{\lambda \in \Lambda} \min_{\gamma \in \Gamma} \left\{ \max_{i \in 1,\ldots,k} \{\lambda_i | z_i^* - \gamma_i | \} \right\}$ where $\gamma \in \Gamma$ corresponds to the set of solutions in a given Pareto front approximations and $z^*$ is the utopian point corresponding to the ideal solution of the MOO. Generally, $R_2$ metric calculations are performed with $z^*$ equal to the origin and all objectives transformed to a minimization setting, which serves to preserve the monotonic nature of the metric. This holds true for our experiments as well.

4. **Top-K Reward** This metric was originally used in (Bengio et al., 2021a), which we extend for our multi-objective setting. For MOGFN-PC, we sample $N$ candidates per test preference and then pick the top-$k$ candidates ($k < N$) with highest scalarized rewards and calculate the mean. We repeat this for all test preferences enumerated from the simplex and report the average top-k reward score.

5. **Top-K Diversity** This metric was also originally used in (Bengio et al., 2021a), which we again extend for our multi-objective setting. We use this metric to quantify the notion of diversity of the generated candidates. Given a distance metric $d(x, y)$ between candidates $x$ and $y$ we calculate the diversity of candidates as those who have $d(x, y)$ greater than a

threshold $\epsilon$. For MOGFN-PC, we sample $N$ candidates per test preference and then pick the top-k candidates based on the diversity scores and take the mean. We repeat this for all test preferences sampled from simplex and report the average top-k diversity score. We use the edit distance for sequences, and 1 minus the Tanimoto similarity for molecules.

# E  ADDITIONAL EXPERIMENTAL DETAILS

## E.1  HYPER-GRID

Here we elaborate on the Hyper-Grid experimental setup which we discussed in Section 5.1.1. Consider an $n$-dimensional hypercube gridworld where each cell in the grid corresponds to a state. The agent starts at the top left coordinate marked as $(0, 0, \dots)$ and is allowed to move only towards the right, down, or stop. When the agent performs the *stop* action, the trajectory terminates and the agent receives a non-zero reward. In this work, we consider the following reward functions - `brannin(x)`, `currin(x)`, `sphere(x)`, `shubert(x)`, `beale(x)`. In Figure 4, we show the heatmap for each reward function. Note that we normalize all the reward functions between 0 and 1.

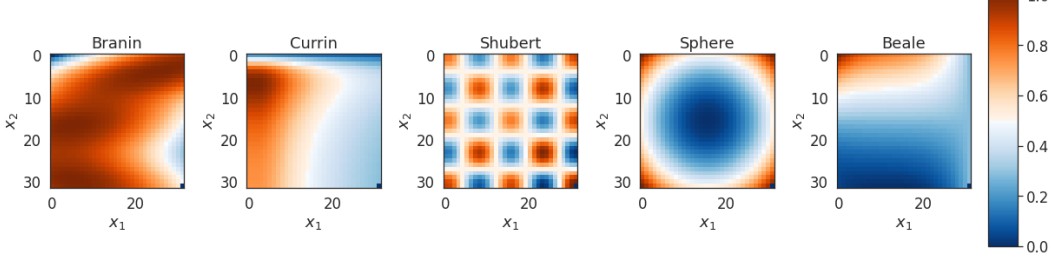

Figure 4: **Reward Functions** Different reward function considered for HyperGrid experiments presented in Section 5.1.1. Here the grid dimension is $H = 32$
.

**Additional Results** To verify the efficacy of MOGFNs across different objectives sizes, we perform some additional experiments and measure the $L_1$ loss and the $GD+$ metric. In Figure 5, we can see that as the reward dimension increases, the loss and $GD+$ increases. This is expected because the number of rewards is indicative of the difficulty of the problem. We also present extended qualitative visualizations across more preferences in Figure 6.

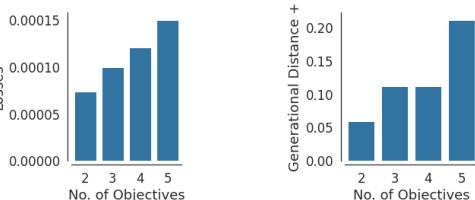

Figure 5: *(Left)* Average test loss between the MOGFN-PC distribution and the true distribution for increasing number of objectives. *(Right)* GD + metrics of MOGFN-PC across objectives.

**Model Details and Hyperparameters** For MOGFN-PC policies we use an MLP with two hidden layers each consisting of 64 units. We use `LeakyReLU` as our activation function as in Bengio et al. (2021a). All models are trained with `learning rate=0.01` with the Adam optimizer Kingma & Ba (2015) and `batch size=128`. We sample preferences $\omega$ from Dirichlet$(\alpha)$ where $\alpha = 1.5$. We try two encoding techniques for encoding preferences - 1) Vanilla encoding where we just use the raw values of the preference vectors and 2) Thermometer encoding (Buckman et al., 2018). In our experiments we have not observed significant difference in performance difference.

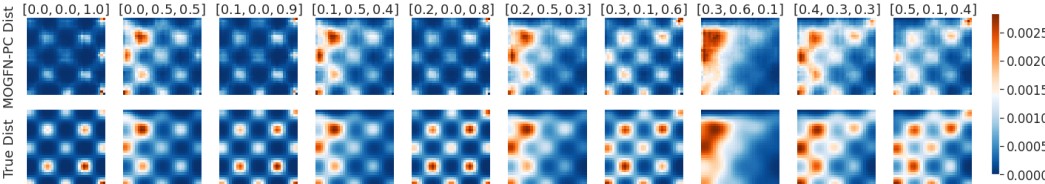

Figure 6: Extended Qualitative Visualizations for Hypergrid epxeriments

## E.2 N-GRAMS TASK

**Task Details** The task is to generate sequences of some maximum length $L$, which we set to 36 for the experiments in Section 5.1.2. We consider a vocabulary (actions) of size 21, with 20 characters ["A", "R", "N", "D", "C", "E", "Q", "G", "H", "I", "L", "K", "M", "F", "P", "S", "T", "W", "Y", "V"] and a special token to indicate the end of sequence. The rewards $\{R_i\}_{i=1}^d$ are defined by the number of occurrences of a given set of n-grams in a sequence $x$. For instance, consider ["AB", "BA"] as the n-grams. The rewards for a sequence $x = $ ABABC would be $[2, 1]$. We consider two choices of n-grams: (a) Unigrams: the number of occurrences of a set of unigrams induces conflicting objectives since we cannot increase the number of occurrences of a monogram without replacing another in a string of a particular length, (b) Bigrams: given common characters within the bigrams, the occurrences of multiple bigrams can be increased simultaneously within a string of a fixed length. We also consider different sizes for the set of n-grams considered, i.e. different number of objectives. This allows us to evaluate the behaviour of MOGFN-PC on a variety of objective spaces. We summarize the specific objectives used in our experiments in Table 8. We normalize the rewards to $[0, 1]$ in our experiments.

Table 8: Objectives considered for the N-grams task

| Objectives | n-grams |
|---|---|
| 2 Unigrams | ["A", "C"] |
| 2 Bigrams | ["AC", "CV"] |
| 3 Unigrams | ["A", "C", "V"] |
| 3 Bigrams | ["AC", "CV", "VA"] |
| 4 Unigrams | ["A", "C", "V", "W"] |
| 4 Bigrams | ["AC", "CV", "VA", "AW"] |

**Model Details and Hyperparameters** We build upon the implementation from Stanton et al. (2022) for the task: https://github.com/samuelstanton/lambo. For the string generation task, the backward policy $P_B$ is trivial (as there is only one parent for each node $s \in \mathcal{S}$), so we only have to parameterize $P_F$ and $\log Z$. As $P_F(-|s, \omega)$ is a conditional policy, we use a Conditional Transformer encoder as the architecture. This consists of a Transformer encoder (Vaswani et al., 2017) with 3 hidden layers of dimension 64 and 8 attention heads to embed the current state (string generated so far) $s$. We have an MLP which embeds the preferences $\omega$ which are encoded using thermometer encoding with 50 bins. The embeddings of the state and preferences are concatenated and passed to a final MLP which generates a categorical distribution over the actions (vocabulary token). We use the same architecture for the baselines using a conditional policy – MOReinforce and MOSoftQL. For Envelope-MOQ, which does not condition on the preferences, we use a standard Transformer-encoder with a similar architecture. We present the hyperparameters we used in Table 9. Each method is trained for 10,000 iterations with a minibatch size of 128. For the baselines we adopt the official implementations released by the authors for MOReinforce – https://github.com/Xi-L/PMOCO and Envelope-MOQ – https://github.com/RunzheYang/MORL.

**Additional Results** We present some additional results for the n-grams task. We consider different number of objectives $d \in \{2, 4\}$ in Table 10 and Table 11 respectively. As with the experiments in Section 5.1.2 we observe that MOGFN-PC outperforms the baselines in Pareto performance while achieving high diversity scores. In Table 12, we consider the case of shorter sequences $L = 24$.

Table 9: Hyperparameters for N-grams Task

| Hyperparameter | Values |
|---|---|
| Learning Rate ($P_F$) | $\{0.01, 0.05, 0.001, 0.005, 0.0001\}$ |
| Learning Rate ($Z$) | $\{0.01, 0.05, 0.001\}$ |
| Reward Exponent: $\beta$ | $\{16, 32, 48\}$ |
| Uniform Policy Mix: $\delta$ | $\{0.01, 0.05, 0.1\}$ |

MOGFN-PC continues to provide significant improvements over the baselines. There are two trends we can observe considering the N-grams task holistically:

1. As the sequence size increases the advantage of MOGFN-PC becomes more significant.
2. The advantage of MOGFN-PC increases with the number of objectives.

Table 10: **N-grams Task**. 2 Objectives

| Algorithm | 2 Bigrams | | | | 2 Unigrams | | | |
|---|---|---|---|---|---|---|---|---|
| | Reward (↑) | Diversity (↑) | HV (↑) | $R_2$ (↓) | Reward (↑) | Diversity (↑) | HV (↑) | $R_2$ (↓) |
| Envelope-MOQ | $0.05_{\pm0.001}$ | $0_{\pm0}$ | $0.0_{\pm0.0}$ | $0_{\pm0.0}$ | $0.09_{\pm0.02}$ | $0_{\pm0}$ | $0.014_{\pm0.001}$ | $5.73_{\pm0.09}$ |
| MOReinforce | $0.12_{\pm0.01}$ | $0_{\pm0}$ | $0.151_{\pm0.023}$ | $0.031_{\pm}$ | $0.43_{\pm0.04}$ | $0_{\pm0}$ | $0.222_{\pm0.013}$ | $2.54_{\pm0.06}$ |
| MOSoftQL | $0.37_{\pm0.03}$ | $19.40_{\pm0.91}$ | $0.247_{\pm0.031}$ | $2.92_{\pm0.39}$ | $0.46_{\pm0.02}$ | $22.05_{\pm0.04}$ | $0.253_{\pm0.003}$ | $2.54_{\pm0.02}$ |
| MOGFN-TB | $0.51_{\pm0.04}$ | $20.65_{\pm0.58}$ | $0.321_{\pm0.011}$ | $2.31_{\pm0.04}$ | $0.48_{\pm0.01}$ | $22.15_{\pm0.22}$ | $0.267_{\pm0.007}$ | $2.24_{\pm0.03}$ |

Table 11: **N-grams Task**. 4 Objectives

| Algorithm | 4 Bigrams | | | | 4 Unigrams | | | |
|---|---|---|---|---|---|---|---|---|
| | Reward (↑) | Diversity (↑) | HV (↑) | $R_2$ (↓) | Reward (↑) | Diversity (↑) | HV (↑) | $R_2$ (↓) |
| Envelope-MOQ | $0_{\pm0}$ | $0_{\pm0}$ | $0_{\pm0}$ | $85.23_{\pm2.78}$ | $0_{\pm0}$ | $0_{\pm0}$ | $0_{\pm0}$ | $80.36_{\pm3.16}$ |
| MOReinforce | $0.01_{\pm0.00}$ | $0_{\pm0}$ | $0.001_{\pm0.001}$ | $60.42_{\pm1.52}$ | $0.00_{\pm0.00}$ | $0_{\pm0}$ | $0_{\pm0}$ | $79.12_{\pm4.21}$ |
| MOSoftQL | $0.12_{\pm0.04}$ | $24.32_{\pm1.21}$ | $0.013_{\pm0.001}$ | $39.31_{\pm1.35}$ | $0.22_{\pm0.02}$ | $24.18_{\pm1.43}$ | $0.019_{\pm0.005}$ | $31.46_{\pm2.32}$ |
| MOGFN-TB | $0.23_{\pm0.02}$ | $20.31_{\pm0.43}$ | $0.055_{\pm0.017}$ | $24.42_{\pm1.44}$ | $0.33_{\pm0.01}$ | $23.24_{\pm0.23}$ | $0.063_{\pm0.032}$ | $23.31_{\pm2.03}$ |

Table 12: **N-grams Task**. Shorter Sequences

| Algorithm | 3 Bigrams | | | | 3 Unigrams | | | |
|---|---|---|---|---|---|---|---|---|
| | Reward ($\uparrow$) | Diversity ($\uparrow$) | HV ($\uparrow$) | $R_2$ ($\downarrow$) | Reward ($\uparrow$) | Diversity ($\uparrow$) | HV ($\uparrow$) | $R_2$ ($\downarrow$) |
| Envelope-MOQ | $0.07_{\pm 0.01}$ | $0_{\pm 0}$ | $0.027_{\pm 0.010}$ | $16.21_{\pm 0.48}$ | $0.08_{\pm 0.02}$ | $0_{\pm 0}$ | $0.031_{\pm 0.015}$ | $20.13_{\pm 0.41}$ |
| MOReinforce | $0.18_{\pm 0.01}$ | $0_{\pm 0}$ | $0.053_{\pm 0.031}$ | $13.35_{\pm 0.82}$ | $0.07_{\pm 0.02}$ | $0_{\pm 0}$ | $0.041_{\pm 0.009}$ | $19.25_{\pm 0.41}$ |
| MOSoftQL | $0.31_{\pm 0.02}$ | $20.12_{\pm 0.51}$ | $0.143_{\pm 0.019}$ | $12.79_{\pm 0.41}$ | $0.38_{\pm 0.02}$ | $21.13_{\pm 0.35}$ | $0.109_{\pm 0.011}$ | $12.12_{\pm 0.24}$ |
| MOGFN-PC | $0.45_{\pm 0.02}$ | $19.62_{\pm 0.04}$ | $0.225_{\pm 0.009}$ | $9.82_{\pm 0.23}$ | $0.39_{\pm 0.01}$ | $21.94_{\pm 0.21}$ | $0.125_{\pm 0.015}$ | $10.91_{\pm 0.14}$ |

## E.3 QM9

**Reward Details** As mentioned in Section 5.2.1, we consider four reward functions for our experiments. The first reward function is the HUMO-LUMO gap, for which we rely on the predictions of a pretrained MXMNet (Zhang et al., 2020) model trained on the QM9 dataset (Ramakrishnan et al., 2014). The second reward is the standard Synthetic Accessibility score which we calculate using the RDKit library (Landrum), to get the reward we compute $(10 - \mathtt{SA})/9$. The third reward function is molecular weight target. Here we first calculate the molecular weight of a molecule using RDKit, and then construct a reward function of the form $e^{-(\mathtt{molWt}-105)^2/150}$ which is maximized at 105. Our final reward function is a logP target, $e^{-(\mathtt{logP}-2.5)^2/2}$, which is again calculated with RDKit and is maximized at 2.5.

**Model Details and Hyperparameters** We sample new preferences for every episode from a $Dirichlet(\alpha)$, and encode the desired sampling temperature using a thermometer encoding (Buckman et al., 2018). We use a graph neural network based on a graph transformer architecture (Yun et al., 2019). We transform this conditional encoding to an embedding using an MLP. The embedding is then fed to the GNN as a virtual node, as well as concatenated with the node embeddings in the graph. The model's action space is to add a new node to the graph, a new bond, or set node or bond properties (like making a bond a double bond). It also has a stop action. For more details please refer to the code provided in the supplementary material. We summarize the hyperparameters used in Table 13.

| Hyperparameter | Value |
|---|---|
| Learning Rate ($P_F$) | 0.0005 |
| Learning Rate ($Z$) | 0.0005 |
| Reward Exponent: $\beta$ | 32 |
| Batch Size: | 64 |
| Number of Embeddings | 64 |
| Uniform Policy Mix: $\delta$ | 0.001 |
| Number of layers | 4 |

Table 13: Hyperparameters for QM9 Task

## E.4 FRAGMENTS

**More Details** As mentioned in Section 5.2.2, we consider four reward functions for our experiments. The first reward function is a proxy trained on molecules docked with AutodockVina (Trott & Olson, 2010) for the sEH target; we use the weights provided by Bengio et al. (2021a). We also use synthetic accessibility, as for QM9, and a weight target *region* (instead of the specific target weight used for QM9), `((300 - molwt) / 700 + 1).clip(0, 1)` which favors molecules with a weight of under 300. Our final reward function is QED which is again calculated with RDKit.

**Model Details and Hyperparameters** We again use a graph neural network based on a graph transformer architecture (Yun et al., 2019). The experimental protocol is similar to QM9 experiments discussed in Appendix E.3. We additionally sample from a lagged model whose parameters are updated as $\theta' = \tau\theta' + (1 - \tau)\theta$. The model's action space is to add a new node, by choosing from a

list of fragments and an attachment point on the current molecular graph. We list all hyperparameters used in Table 14.

| Hyperparameter | Value |
|---|---|
| Learning Rate ($P_F$) | 0.0005 |
| Learning Rate ($Z$) | 0.0005 |
| Reward Exponent: $\beta$ | 96 |
| Batch Size: | 256 |
| Sampling model $\tau$ | 0.95 |
| Number of Embeddings | 128 |
| Number of layers | 6 |

Table 14: Hyperparameters for Fragments

**Additional Results** We also present in Figure 7 a view of the reward distribution produced by MOGFN-PC. Generally, the model is able to find good near-Pareto-optimal samples, but is also able to spend a lot of time exploring. The figure also shows that the model is able to respect the preference conditioning, and remains capable of generating a diverse distribution rather than a single point.

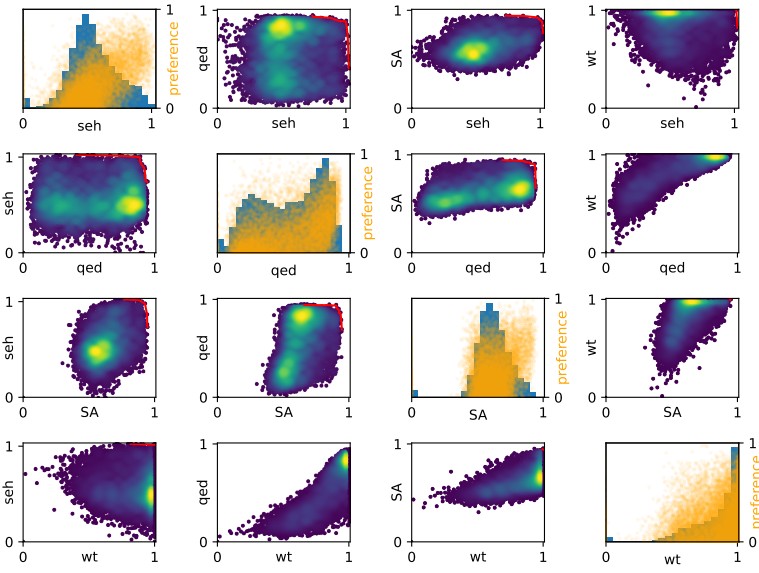

Figure 7: Fragment-based molecule generation: See Appendix E.4.

In the off-diagonal plots of Figure 7, we show pairwise scatter plots for each objective pair; the Pareto front is depicted with a red line; each point corresponds to a molecule generated by the model as it explores the state space; color is density (linear viridis palette). The diagonal plots show two overlaid informations: a blue histogram for each objective, and an orange scatter plot showing the relationship between preference conditioning and generated molecules. The effect of this conditioning is particularly visible for `seh` (top left) and `wt` (bottom right). As the preference for the sEH binding reward gets closer to 1, the generated molecules' reward for sEH gets closer to 1 as well. Indeed, the expected shape for such a scatter plot is a triangular-ish shape: when the preference $\omega_i$ for reward $R_i$ is close to 1, the model is expected to generate objects with a high reward for $R_i$; as the preference $\omega_i$ gets further away from 1, the model can generate anything, including objects with a high $R_i$–that is, unless there is a trade off between objectives, in which case in cannot; this is the case for the `seh` objective, but not for the `wt` objective, which has a more triangular shape.

## E.5 DNA Sequence Design

**Task Details** The set of building blocks here consists of the bases [`"A"`, `"C"`, `"T"`, `"G"`] in addition to a special end of sequence token. In order to compute the free energy and number of base with the software NUPACK (Zadeh et al., 2011), we used 310 K as the temperature. The inverse of the length $L$ objective was calculated as $\frac{30}{L}$, as 30 was the minimum length for sampled sequences. The rewards are normalized to $[0, 1]$ for our experiments.

**Model Details and Hyperparameters** We use the same implementation as the N-grams task, detailed in Appendix E.2. Here we consider a 4-layer Transformer architecture, with 256 units per layer and 16 attention head instead. We detail the most relevant hyperparameters Table 15.

Table 15: Hyperparameters tuned for DNA-Aptamers Task.

| Hyperparameter | Values |
|---|---|
| Learning Rate ($P_F$) | {0.001, 0.0001, 0.00001, 0.000001} |
| Learning Rate ($Z$) | 0.001 |
| Reward Exponent: $\beta$ | {40, 60, 80} |
| Batch Size: | 16 |
| Training iterations: | 10,000 |
| Dirichlet $\alpha$ | {0.1, 1.0, 1.5} |

**Discussion of Results** Contrary to the other tasks on which we evaluated MOGFN-PC, for the generation of DNA aptamer sequences, our proposed model did not match the best baseline, multi-objective reinforcement learning (Lin et al., 2021), in terms of Pareto performance. Nonetheless, it is worth delving into the details in order to better understand the different solutions found by the two methods. First, as indicated in section 5, despite the better Pareto performance, the best sequences generated by the RL method have extremely low diversity (0.62), compared to MOGFN, which generates optimal sequences with diversity of 19.6 or higher. As a matter of fact, MOReinforce mostly samples sequences with the well-known pattern GCGC... for all possible lengths. Sequences with this pattern have indeed low (negative) energy and many number of pairs, but they offer little new insights and poor diversity if the model is not able to generate sequences with other distinct patterns. On the contrary, GFlowNets are able to generate sequences with patterns other than repeating the pair of bases G and C. Interestingly, we observed that GFlowNets were able to generate sequences with even lower energy than the best sequences generated by MOReinforce by inserting bases A and T into chains of GCGC.... Finally, we observed that one reason why MOGFN does not match the Pareto performance of MOReinforce is because for short lengths (one of the objectives) the energy and number of pairs are not successfully optimised. Nonetheless, the optimisation of energy and number of pairs is very good for the longest sequences. Given these observations, we conjecture that there is room for improving the set of hyperparameters or certain aspects of the algorithm.

**Additional Results** In order to better understand the impact of the main hyperparameters of MOGFN-PC in the Pareto performance and diversity of the optimal candidates, we train multiple instances by sweeping over several values of the hyperparameters, as indicated in Table 15. We present the results in Table 16. One key observation is that there seems to be a tradeoff between the Pareto performance and the diversity of the Top-K sequences. Nonetheless, even the models with the lowest diversity are able to generate much more diverse sequences than MOReinforce. Furthermore, we also observe $\alpha < 1$ as the parameter of the Dirichlet distribution to sample the weight preferences, as well as higher $\beta$ (reward exponent), both yield better metrics of Pareto performance but slightly worse diversity. In the case of $\beta$, this observation is consistent with the results in the Bigrams task (Table 5), but with Bigrams, best performance was obtained with $\alpha = 1$. This is indicative of a degree of dependence on the task and the nature of the objectives.

## E.6 Active Learning

**Task Details** We consider the Proxy RFP task from Stanton et al. (2022), an in silico benchmark task designed to simulate searching for improved red fluorescent protein (RFP) variants (Dance et al., 2021). The objectives considered are stability (-dG or negative change in Gibbs free energy) and

Table 16: Analysis of the impact of $\alpha$, $\beta$ and the learning rate on the performance of MOGFN-PC for DNA sequence design. We observe a trade-off between the Top-K diversity and the Pareto performance.

| Metrics | Effect of $p(\omega)$ | | | Effect of $\beta$ | | | Effect of the learning rate | | | |
|---|---|---|---|---|---|---|---|---|---|---|
| | Dir($\alpha=0.1$) | Dir($\alpha=1$) | Dir($\alpha=1.5$) | 40 | 60 | 80 | $10^{-5}$ | $10^{-4}$ | $10^{-3}$ | $10^{-2}$ |
| Reward ($\uparrow$) | $0.687_{\pm0.01}$ | $0.652_{\pm0.01}$ | $0.639_{\pm0.01}$ | $0.506_{\pm0.01}$ | $0.560_{\pm0.01}$ | $0.652_{\pm0.01}$ | $0.587_{\pm0.01}$ | $0.652_{\pm0.01}$ | $0.654_{\pm0.03}$ | $0.604_{\pm0.01}$ |
| Diversity ($\uparrow$) | $17.65_{\pm0.37}$ | $19.58_{\pm0.15}$ | $20.18_{\pm0.58}$ | $28.49_{\pm0.32}$ | $24.93_{\pm0.19}$ | $19.58_{\pm0.15}$ | $21.92_{\pm0.59}$ | $19.58_{\pm0.15}$ | $19.51_{\pm1.14}$ | $23.16_{\pm0.18}$ |
| Hypervolume ($\uparrow$) | $0.506_{\pm0.01}$ | $0.467_{\pm0.02}$ | $0.440_{\pm0.01}$ | $0.277_{\pm0.03}$ | $0.363_{\pm0.03}$ | $0.467_{\pm0.02}$ | $0.333_{\pm0.01}$ | $0.467_{\pm0.02}$ | $0.496_{\pm0.01}$ | $0.336_{\pm0.01}$ |
| $R_2$ ($\downarrow$) | $2.462_{\pm0.05}$ | $2.576_{\pm0.08}$ | $2.688_{\pm0.02}$ | $4.225_{\pm0.34}$ | $2.905_{\pm0.18}$ | $2.576_{\pm0.08}$ | $3.855_{\pm0.31}$ | $2.576_{\pm0.01}$ | $2.488_{\pm0.03}$ | $3.422_{\pm0.07}$ |

solvent-accessible surface area (SASA) (Shrake & Rupley, 1973) in simulation, computed using the FoldX suite (Schymkowitz et al., 2005) and BioPython (Cock et al., 2009). We use the dataset introduced in Stanton et al. (2022) as the initial pool of candidates $\mathcal{D}_0$ with $|\mathcal{D}_0| = 512$.

**Method Details and Hyperparameters** Our implementation builds upon the publicly released code from (Stanton et al., 2022): `https://github.com/samuelstanton/lambo`. We follow the exact experimental setup used in Stanton et al. (2022). The surrogate model $\hat{f}$ consists of an encoder with 1D convolutions (masking positions corresponding to padding tokens). We used 3 standard pre-activation residual blocks with two convolution layers, layer norm, and swish activations, with a kernel size of 5, 64 intermediate channels and 16 latent channels. A multi-task GP with an ICM kernel is defined in the latent space of this encoder, which outputs the predictions for each objective. We also use the training tricks detailed in Stanton et al. (2022) for the surrogate model. The hyperparameters, taken from Stanton et al. (2022) are shown in Table 17. The acquisiton function used is NEHVI (Daulton et al., 2021) defined as

$$\alpha(\{x_j\}_{j=1}^i) = \frac{1}{N}\sum_{t=1}^{N}\mathrm{HVI}(\{\tilde{f}_t(x_j)\}_{j=1}^{i-1}|\mathcal{P}_t) + \frac{1}{N}\sum_{t=1}^{N}\mathrm{HVI}(\tilde{f}_t(x_j)|\mathcal{P}_t \cup \{\tilde{f}_t(x_j)\}_{j=1}^{i-1}) \quad (3)$$

where $\tilde{f}_t, t = 1, \dots N$ are independent draws from the surrogate model (which is a posterior over functions), and $\mathcal{P}_t$ denotes the Pareto frontier in the current dataset $\mathcal{D}$ under $\tilde{f}_t$.

Table 17: Hyperparameters for training the surrogate model $\hat{f}$

| Hyperparameter | Value |
|---|---|
| Shared enc. depth (# residual blocks) | 3 |
| Disc. enc. depth (# residual blocks) | 1 |
| Decoder depth (# residual blocks) | 3 |
| Conv. kernel width (# tokens) | 5 |
| # conv. channels | 64 |
| Latent dimension | 16 |
| GP likelihood variance init | 0.25 |
| GP lengthscale prior | N(0.7, 0.01) |
| # inducing points (SVGP head) | 64 |
| DAE corruption ratio (training) | 0.125 |
| DAE learning rate (MTGP head) | 5.00E-03 |
| DAE learning rate (SVGP head) | 1.00E-03 |
| DAE weight decay | 1.00E-04 |
| Adam EMA params | 0., 1e-2 |
| Early stopping holdout ratio | 0.1 |
| Early stopping relative tolerance | 1.00E-03 |
| Early stopping patience (# epochs) | 32 |
| Max # training epochs | 256 |

We replace the LaMBO candidate generation with GFlowNets. We generate a set of mutations $m = \{(l_i, v_i)\}$ for a sequences $x$ from the current approximation of the Pareto front $\hat{\mathcal{P}}_i$. Note

that, as opposed to the sequence generation experiments, $P_B$ here is not trivial as there are multiple ways (orders) of generating the set. For our experiments, we use a uniform random $P_B$. $P_F$ takes as input the sequence $x$ with the mutations generated so far applied. We use a Transformer encoder with 3 layers, with hidden dimension 64 and 8 attention heads as the architecture for the policy. The policy outputs a distribution over the locations in $x$, $\{1, \ldots, |x|\}$, and a distribution over tokens for each location. The vocabulary of actions here is the same as the N-grams task - ["A", "R", "N", "D", "C", "E", "Q", "G", "H", "I", "L", "K", "M", "F", "P", "S", "T", "W", "Y", "V"]. The logits of the locations of the mutations generated so far are set to -1000, to prevent generating the same sequence. The acquisition function(NEHVI) value for the mutated sequence is used as the reward. We also use a reward exponent $\beta$. To make optimization easier (as the acquisition function becomes harder to optimize with growing $\beta$), we reduce $\beta$ linearly by a factor $\delta\beta$ at each round. We train the GFlowNet for 750 iterations in each round. Table 18 shows the MOGFN-AL hyperparameters. The active learning batch size is 16, and we run 64 rounds of optimization. Table 18 presents the hyperparameters used for MOGFN-AL.

Table 18: Hyperparameters for MOGFN-AL

| Hyperparameter | Values |
|---|---|
| Learning Rate ($P_F$) | $\{0.01, 0.001, 0.0001\}$ |
| Learning Rate ($Z$) | $\{0.01, 0.001\}$ |
| Reward Exponent: $\beta$ | $\{16, 24\}$ |
| Uniform Policy Mix: $\delta$ | $\{0.01, 0.05\}$ |
| Maximum number of mutations | $\{10, 15, 20\}$ |
| $\delta\beta$ | $\{0.5, 1, 2\}$ |

