# OpenReview forum: "Multi-Objective GFlowNets"
_ICLR.cc/2023/Conference — Submitted to ICLR 2023_

### Official Review · Reviewer_Qeid · 2022-10-24

**Confidence:** 3
**Correctness:** 2
**Technical Novelty And Significance:** 2
**Empirical Novelty And Significance:** 2
**Recommendation:** 3

**Clarity, Quality, Novelty And Reproducibility:**

This paper is not well-organized. The problem to be solved is unclear and the novelty of the proposed method needs more discussions.

**Strength And Weaknesses:**

Strengths
1.	This paper extends the GFlowNets proposed for single-objective optimization to multi-objective optimization.
2.	Analyses are conducted to study the main components of MOGFNs.

Weaknesses
1.	The goal of the research in this paper is not clear. How is diversity defined if it is not to cover the Pareto front of a multi-objective optimization problem?
2.	The proposed algorithm is mainly based on the assumption that a multi-objective optimization problem can be viewed as a family of single-objective problems defined by non-negative weighted vectors. This might be inappropriate when the Pareto front is not convex.
3.	Similar to this paper, decomposition-based multi-objective evolutionary algorithms such as MOEA/D solve multi-objective optimization problems by considering a set of single-objective sub-problems. The advantages of the proposed method over such methods need to be discussed and evaluated.


**Summary Of The Paper:**

This paper extends the GFlowNets proposed for single-objective optimization to multi-objective optimization, namely MOGFNs. Experiments are conducted on molecule generation and sequence generation to study the effectiveness of the proposed method.

**Summary Of The Review:**

This paper proposes a new method for multi-objective optimization. However, the advantages and effectiveness of the proposed method are not well described and evaluated.

---

> ### Author Response · Authors · 2022-11-08
> **Response to Reviewer Qeid**
>
> We thank the reviewer for the thoughtful comments and feedback!
>
> > 1. The goal of the research in this paper is not clear. How is diversity defined if it is not to cover the Pareto front of a multi-objective optimization problem?
> >
>
> Coverage of the Pareto front is a widely studied measure of diversity in MOO. In our work, we consider an additional notion of diversity in the input space at each point on the Pareto front which is important for applications like drug discovery. For example, consider the n-gram task in Section 5.1.2 where given a set of symbols (alphabet) we would like to construct sequences with a maximum length of 3 which optimize two rewards - [number of “C”s, number of “A”s]. In this case, (2,1) will be a point on the Pareto front. However there are multiple candidates in the input space which result in this same point in objective space - ACC, CAC, CCA. Our notion of diversity in the input space captures the goal of discovering each of these candidates and being able to sample any of them from the GFlowNet’s generative policy. As we motivate in the introduction, this can be useful in the scientific discovery setting where the rewards can be underspecified, i.e., the true reward is not something that we can compute (e.g. a clinical trial, in drug discovery) and we rely on a proxy that approximates it imperfectly (e.g. in-vitro experiments). In this setting, we desire high diversity in the set of generated candidates, thereby maximizing the chances that some candidate yields a high proxy value while also achieving a high true reward.
>
> > 2. The proposed algorithm is mainly based on the assumption that a multi-objective optimization problem can be viewed as a family of single-objective problems defined by non-negative weighted vectors. This might be inappropriate when the Pareto front is not convex.
> >
>
> Scalarization of MOO problems is a well studied topic in the literature [1, 2, 3] and, with the right choice of scalarization function, it can tackle a wide range of MOO problems. Concretely, any Pareto optimal solution can be found by solving the weighted Tchebycheff scalarization, which we consider, with appropriate parameters. Moreover, for any set of parameters its solution corresponds to a weakly Pareto optimal solution of the original problem [3]. We discuss the use of weighted Tchebycheff scalarization in Section 3.1 and evaluate it empirically in Section 6. However, in our evaluations we found that it performed poorly on the tasks we consider. See also our response to reviewer MZeU on this point.
>
> > 3. Similar to this paper, decomposition-based multi-objective evolutionary algorithms such as MOEA/D solve multi-objective optimization problems by considering a set of single-objective sub-problems. The advantages of the proposed method over such methods need to be discussed and evaluated.
> >
>
> Our approach differs from MOEA/D and similar evolutionary algorithms in two key ways as we discuss in Section 3.1 and 4:
>
> 1. Evolutionary algorithms do not leverage data from past experience, resulting in poor sample efficiency.
> 2. Evolutionary algorithms do not capture the candidate space diversity.
>
> Note that we also consider EA-based baselines in our experiments in Section 5.3.
>
> > This paper is not well-organized. The problem to be solved is unclear and the novelty of the proposed method needs more discussions.
> >
>
> Based on reviewers’ feedback, we have uploaded a revision that aims to clarify the overall goals of the work and the questions raised by the reviewers. We hope along with our responses the changes in the paper will address your concerns.
>
> If you have any additional questions, comments or feedback that would help to alleviate your concerns we would be happy to address them!
>
> [1] “Multicriteria optimization”, Ehrgott, 2005.
>
> [2] “Nonlinear multiobjective optimization”, Miettinen, 2012
>
> [3] “Non-Convex Multi-Objective Optimization”, Pardalos, Žilinskas, Žilinskas. 2017.

---

> ### Author Response · Authors · 2022-11-14
> **Discussion and additional questions**
>
> We thank you once again for the feedback and comments on the paper. As the discussion period comes to an end in a few days, we believe we have addressed your concerns regarding the problem setup and the notion of diversity through our response and changes to Section 1 and Section 2.1,  as well as clarified the generality of our scalarization based approach and differences compared to existing MOEA/D approaches. We encourage you to consider our responses in your evaluation, and we would be happy to answer any further questions you might have!

---

> ### Author Response · Authors · 2022-12-08
> **Discussion and Response to Rebuttal**
>
> Dear Reviewer Qeid,
>
> We thank you again for your insightful review! We appreciate the points you raise regarding non-convex Pareto front and the motivation for our proposed notion of diversity. We have addressed these issues in our updated draft as well as our rebuttal. Your feedback has helped us improve the paper. We would appreciate it if you could take the updated draft and our rebuttal into consideration in your evaluation of the paper. We are also happy to answer any further questions you might have before the end of the discussion period!

---

### Official Review · Reviewer_sp4N · 2022-10-25

**Confidence:** 2
**Correctness:** 3
**Technical Novelty And Significance:** 3
**Empirical Novelty And Significance:** 3
**Recommendation:** 8

**Clarity, Quality, Novelty And Reproducibility:**

The technical novelty of the paper is how to apply and generalize the previously derived method to the MOO problem. I believe this technical novelty is solid. However, the novelty in terms of new ideas seems incremental to me. The paper is well-written with some exceptions that I mentioned. I believe the results are reproducible.

**Strength And Weaknesses:**

Strength

* The authors successfully applied conditional GFlowNet (for the first time) to the Multi-Objective Optimization problem.
* The author generalizes the GFlowNet active learning algorithm for the Multi-Objective Optimization problem case.
* The results seem promising for both algorithms in terms of key metrics and especially the diversity of the generated candidate.
* The results and comparisons with other methods are well explained. (e.g. why in some cases other baselines have higher scores on some of the metrics)

Weaknesses
* I believe the novelty is limited in terms of new ideas as the main technical contribution of the paper is the adaptation of the previously derived method to MOO setting.

Minor issues and questions
* To make the paper self-consistent can you add how \alpha function in the Active Learning section combines a reward with an epistemic uncertainty?
* It is unclear to me from the paper how \alpha function map a set of rewards and their epistemic uncertainties to a single objective in each round of the Active Learning pipeline. Or is the \alpha function defined externally for each round and can be considered as a hyperparameter?
* Do you retrain GFlowNet from scratch for each round of Active Learning?

**Summary Of The Paper:**

The paper leverages GFlowNet to solve the Multi-Objective Optimization problem. The authors derive two versions of the algorithm, regular and active learning variants. The authors empirically demonstrate that the proposed algorithms outperform existing methods almost on every considered benchmark and perform reasonably on DNA sequence design tasks. The method as expected performs very well in terms of generating a diverse set of candidates.

**Summary Of The Review:**

In general, I believe the novelty of the paper is incremental as it is a straightforward adaptation of  GFlowNets to MOO settings. However, this adaptation may possess a solid technical contribution.

---

> ### Author Response · Authors · 2022-11-08
> **Response to Reviewer sp4N**
>
> We thank the reviewer for the thoughtful comments and feedback!
>
> > I believe the novelty is limited in terms of new ideas as the main technical contribution of the paper is the adaptation of the previously derived method to MOO setting.
> >
>
> As we mention in our overall response, we believe our technical contributions are are quite substantial. Our work is the first to demonstrate using GFlowNets for modelling a family of rewards *simultaneously.* We believe this is a critical contribution, which enables GFlowNets to tackle a much larger set of problems and serves as a basis for future work. Within the context of MOO, our work presents a novel and important perspective on diversity. Traditionally, the notion of diversity studied in MOO has been the coverage of the Pareto front. However, as we discuss in Section 1, for practical settings such as drug discovery, it is important to discover diverse candidates *in candidate space* corresponding to each point on the Pareto front *in output space*. Combining these ideas enables GFlowNets to successfully tackle practically relevant scientific discovery tasks such as molecule generation and protein design. We believe these contributions are novel and substantial, and supported by significant improvements in empirical performance relative to baselines.
>
> > To make the paper self-consistent can you add how \alpha function in the Active Learning section combines a reward with an epistemic uncertainty?
> >
>
> We have added the definition for the acquisition function we use - Noisy Expected Hypervolume Improvement [1] - to Appendix E.6, Equation (3) on Page 22.
>
> We would like to emphasize that our method is not limited to this acquisition function and can incorporate other MO acquisition functions like MESMO [2]. We use NEHVI to be consistent with prior work and baselines like LaMBO. We use the implementation from BoTorch [3] to compute this acquisition function.
>
> > It is unclear to me from the paper how \alpha function map a set of rewards and their epistemic uncertainties to a single objective in each round of the Active Learning pipeline. Or is the \alpha function defined externally for each round and can be considered as a hyperparameter?
> >
>
> The function $\alpha$ is the acquisition function (a term used in Bayesian Optimization, BO). We leverage work from the vast multi-objective BO literature. This acquisition function has a fixed form, and takes as input the surrogate model $\hat{f}$ along with the candidate $x$. The particular acquisition function we use computes the expected hypervolume improvement by incorporating the point $x$ given $\hat{f}$. This amounts to taking samples $\tilde{f}$ from $\hat{f}$ (which represents a posterior over functions) and computing the hypervolume improvement under the sampled $f$ upon adding $x$, and averaging over samples of $f$. The hypervolume improvement is computed using box decomposition algorithms in BoTorch. [3]
>
> > Do you retrain GFlowNet from scratch for each round of Active Learning?
> >
>
> That is a great question! We did try both - retraining from scratch and starting from the previous checkpoints and in our experiments we found both perform similarly.
>
> > The technical novelty of the paper is how to apply and generalize the previously derived method to the MOO problem. I believe this technical novelty is solid. However, the novelty in terms of new ideas seems incremental to me.
> >
>
> We hope that our general response and the response to your review addresses your concerns about novelty. We are happy to address any additional feedback or questions you might have to alleviate your concerns!
>
> [1] “Parallel Bayesian Optimization of Multiple Noisy Objectives with Expected Hypervolume Improvement”, Daulton, Balandat, Bakshy, 2020
>
> [2] “Max-value entropy search for multi-objective bayesian optimization”, Belakaria, Deshwal, Doppa, 2019.
>
> [3] "BoTorch: A Framework for Efficient Monte-Carlo Bayesian Optimization", Balandat et al. 2020

---

> > ### Comment · Reviewer_sp4N · 2022-12-02
> > **response to Authors**
> >
> > Dear Authors, thank you for the detailed feedback and I am sorry for my delayed answer! I read your responses and other reviews. Your response partly addressed my concern about novelty. Plus, I think your responses addressed other reviews' concerns (convexity assumption) despite missing reviewers' *MZeU* and *Qeid* replies (hope they can chime in though). So I raised my score to Accept, but with low confidence.

---

> > > ### Author Response · Authors · 2022-12-02
> > > **Thank you for the response**
> > >
> > > Thank you for considering our responses in your evaluation! We are also happy to address any additional questions or concerns you might have!

---

> ### Author Response · Authors · 2022-11-14
> **Discussion and additional questions**
>
> We thank you once again for the feedback and comments on the paper. As the discussion period comes to an end in a few days, we believe we have addressed your major concern regarding the novelty of our framework in our response and answered your questions regarding the active learning setting. We encourage you to consider our responses in your evaluation, and we would be happy to answer any further questions you might have!

---

### Official Review · Reviewer_MZeU · 2022-10-28

**Confidence:** 4
**Correctness:** 3
**Technical Novelty And Significance:** 2
**Empirical Novelty And Significance:** 2
**Recommendation:** 3

**Clarity, Quality, Novelty And Reproducibility:**

Clarity: The presentation is good. But at the knowledge level, this paper does not study the background of MOO well.

Quality: Fairly good.

Novelty: The novelty is limited from the technical level, as weighted sum scalarization seems too simple in MOO. But it seems good from the application level with respect to drug discovery

Reproducibility: Good.


**Strength And Weaknesses:**

Strengths:
+ The idea of extending gflow nets to the multi-objective setting is novel and interesting.
+ Multi-objective gflow nets are useful in drug discovery.

Weaknesses:
+ The extension of weighted sum scalarization to glow nets seems incremental, which lacks novelty.
+ The understanding of multi-objective of this paper is limited. Some of the assertions are overclaimed. For example, weighted sum scalarization is only one simple way to do multi-objective optimization (MOO). It has many drawbacks, especially when non-convex Pareto front is encountered. But in Section 2.1, the authors claim solving weighted sum scalarization equals solving MOO. More precise background knowledge on MOO should be incorporated.
+ The study of the active version of gflow nets is not very clearly motivated.




**Summary Of The Paper:**

This paper extends the simple linear scalarization method in multi-objective optimization to the GflowNets setting. Two variants of GflowNets are proposed and extensive empirical experiments are provided, which verifies the effectiveness of them.

**Summary Of The Review:**

More technical depth of MOO should be incorporated. The technical contribution of extending MOO to GlowNets seems to be limited. The motivation of the study should be more precisely described.

---

> ### Author Response · Authors · 2022-11-08
> **Response to Reviewer MZeU (1/2)**
>
> We thank the reviewer for the thoughtful comments and feedback!
>
> > The extension of weighted sum scalarization to glow nets seems incremental, which lacks novelty.
> >
>
> In addition to our general response, We would like to clarify that our contribution is not merely using weighted sum scalarization with GFlowNets. Our work is the first one to successfully leverage reward-conditional GFlowNets for modeling a family of reward functions simultaneously. In fact, as we discuss in Section 3.1 and Section 6, our method is not constrained to weighted sum scalarization but can use any scalarization scheme as a drop-in replacement. To demonstrate this we consider two additional scalarization schemes - Weighted-Tchebycheff and Weighted-log-sum, where notably Weighted-Tchebycheff is designed for non-convex Pareto fronts. We further support this with a series of ablations experiments on the n-gram task and study the Pareto performance and diversity under different scalarizations. In addition to the ablations on the string task, we will also add a similar ablation for the fragments task. We believe the contributions are novel and substantial.
>
> > The understanding of multi-objective of this paper is limited. Some of the assertions are overclaimed. For example, weighted sum scalarization is only one simple way to do multi-objective optimization (MOO). It has many drawbacks, especially when non-convex Pareto front is encountered. But in Section 2.1, the authors claim solving weighted sum scalarization equals solving MOO. More precise background knowledge on MOO should be incorporated.
> >
>
> We agree that weighted sum scalarization is a simple way to decompose the MOO problem, and as we state in Section 2.1 the correspondence holds only *under certain assumptions* - which as you rightly point out is the convexity of the Pareto front. But as we highlight above, our method is not limited to weighted sum scalarization. Our method can use any scalarization scheme as detailed in Section 3.1 and empirically validated in Section 6. In fact, in our experiments we observed that Weighted-Tchebycheff performs poorly, possibly due to the difficult optimization landscape. We have reworked Section 2.1 to provide a broader overview of scalarization methods in MOO. We would be happy to incorporate key references we might have missed. See also our response to reviewer Qeid on this point.
>
> > The study of the active version of gflow nets is not very clearly motivated.
> >
>
> In many practical settings, the reward functions we are interested in can be very computationally expensive. For instance, the FoldX simulator we use in our active learning experiments can take several hours to process a single sequence. In these cases we would like to find Pareto optimal candidates in the fewest possible calls to the reward functions. Such a setting motivates the active learning setup for MOGFN-AL used in the paper. We have tried to emphasize this motivation clearly in section 3.2 in our updated draft.
>
> > Clarity: The presentation is good. But at the knowledge level, this paper does not study the background of MOO well.
> >
>
> As mentioned above, we have expanded the scope of our discussion on MOO in the updated draft. We are happy to to incorporate any additional suggestions you might have to improve the discussion.
>
> > Novelty: The novelty is limited from the technical level, as weighted sum scalarization seems too simple in MOO. But it seems good from the application level with respect to drug discovery
> >
>
> We would like to emphasize that our work is the first to demonstrate the use of GFlowNets for modeling a family of rewards *simultaneously.* We believe this is a critical contribution of our work and can serve as a basis for future directions. Within the context of MOO, our work presents a novel perspective on diversity in the candidate space. Traditionally, the notion of diversity studied in MOO has been the coverage of the Pareto front in the solution space. However, as we discuss in Section 1, for practical settings such as drug discovery, it is important to discover diverse candidates (in candidate space) corresponding to each point on the Pareto front (in output space). In our opinion, this is a truly novel contribution that expands upon common considerations and capabilities in MOO. As we note above, our framework is general and can build upon existing scalarization-based MOO approaches and is not limited to weighted sum scalarization. We believe these contributions are substantial, as supported by the significant empirical improvements shown in experiments.

---

> > ### Author Response · Authors · 2022-11-08
> > **Response to Reviewer MZeU (2/2)**
> >
> >
> > > More technical depth of MOO should be incorporated. The technical contribution of extending MOO to GlowNets seems to be limited. The motivation of the study should be more precisely described.
> > >
> >
> > We provided some additional motivation in our general comments, and as mentioned earlier we have reworked Section 2.1 to emphasize the general scalarization approach. Regarding novelty please see the response above.
> >
> > Please let us know if you have any additional feedback or questions that we can address to alleviate your concerns.

---

> ### Author Response · Authors · 2022-11-14
> **Discussion and additional questions**
>
> We thank you once again for the feedback and comments on the paper. As the discussion period comes to an end in a few days, we believe we have addressed  your major concerns regarding the background on MOO with the re-written Section 2.1 and clarified the novelty and flexibility of our framework to incorporate any scalarization, as well as the primary motivations for the active learning setting in our responses here. We encourage you to consider our responses in your evaluation, and we would be happy to answer any further questions you might have!

---

> ### Author Response · Authors · 2022-12-08
> **Discussion and Response to Rebuttal**
>
> Dear Reviewer MZeU,
>
> We thank you again for your insightful review! We appreciate the points you raise regarding the exposition of multi-objective optimization and the active learning setting. We have addressed these issues in our updated draft, and also made it clear that the method is not limited to weighted sum scalarization and works with any scalarization scheme. Your feedback has helped us make the paper much clearer. We would appreciate it if you could take the updated draft and our rebuttal into consideration in your evaluation of the paper. We are also happy to answer any further questions you might have before the end of the discussion period!

---

### Author Response · Authors · 2022-11-08
**Overall Response**

We thank all the reviewers for the thoughtful comments and feedback. We respond to each reviewer individually and address some broader concerns here. We hope that our response addresses all the concerns raised by the reviewers.

---

We believe our work presents novel technical and conceptual contributions including the first successful instantiation of a conditional GFlowNet, as well as a novel perspective of diversity to tackle a broad range of practically relevant multi-objective optimization problems, resulting in **diverse Pareto-optimal candidates** for the same Pareto-optimal solution.

We expand a bit on some of these aspects:

1. **Motivation:** Our work is motivated primarily by practical scientific discovery applications, such as the case of drug discovery we describe in the introduction. The objectives optimized by MOO algorithms are often *underspecified,* as it is often not possible to capture the true underlying mechanisms. This introduces uncertainty in the objectives, making diversity in candidate space critical. This notion of diversity is complementary to the diversity in objective space (Pareto front coverage) that is often considered in MOO. For instance, candidate diversity increases the likelihood of success of wet-lab experiments in drug discovery as it reduces the impact of *local errors* that models of objectives inevitably make when extrapolating.
2. **Novelty:** We would like to highlight that our work contains several novel technical contributions.
    1. **Diversity in Candidate-Space**: We present a novel perspective on diversity in multi-objective optimization motivated by practical scientific discovery scenarios. Specifically, with our notion of candidate-space diversity we are interested in discovering all the candidates corresponding to each point on the Pareto front. As we try to motivate above, ensuring this notion of diversity in candidate space can improve the likelihood of finding useful candidates in practice. We believe this perspective will foster novel directions for MOO.
    2. **Extending GFlowNets for Multi-Objective Optimization**: To capture this diversity, our work extends GFlowNets for multi-objective optimization. We also present the first successful instantiation of **reward-conditional** GFlowNets. Through our experiments, we test our method on a wide variety of challenging high-dimensional, multi-objective tasks that are practically relevant from molecule generation to protein design. Our empirical results demonstrate that MOGFNs can discover better Pareto fronts while generating diverse candidates, enabling a wide-variety of important applications, such as molecule generation and protein design. We believe these contributions are substantial in the context of GFlowNets as well as scientific applications.
3. **Assumption of convexity:** As the reviewers pointed out, Section 2.1 only discusses weighted sum scalarization, which is limited to problems with a convex Pareto front. While not highlighted clearly in the submitted version of the paper, we would like to emphasize that our method is quite general and can accommodate **any scalarization method**, including ones designed for non-convex Pareto front. As described in Section 3.1, we already considered other choices for scalarization - notably Weighted-Tchebycheff which is applicable for non-convex Pareto fronts. We also present empirical results with Weighted-Tchebycheff scalarization in Section 6. We have emphasized this aspect more in our updated draft.

Based on the reviewers feedback, we have made the following changes to our draft (changes in the PDF are colored red):

1. Improved the background on Multi-Objective Optimization considering the general scalarization approach and highlighting the flexibility of MOGFN to use any scalarization function.
2. Added discussion about the acquisition function used in Section 3.2 in Appendix E.6.
3. Changes in introduction to emphasize the importance of diversity in the context of scientific discovery.

Additionally, we will also be adding results for an ablation study similar to Section 6 for the molecule generation task, studying the choice of scalarization, $\beta$ and $p(\omega)$, in a few days.

---

### Author Response · Authors · 2022-11-12
**Additional Experiments**

To provide further details on the capabilities and adaptability of MOGFN-PC, we have updated the paper with additional results for ablations on the fragment-based molecule generation task (Section 5.2.2), in Table 6 studying the effect of $\beta$, $p(\omega)$ and the scalarization $R(x|\omega)$. As we discuss in Section 6, we observe similar trends for fragment-based molecule generation as in the n-grams task, specifically higher $\beta$ leads to better Pareto performance, and weighted sum scalarization performs best, potentially due to easier optimization landscape for the policy. We would like to note that this molecule task is quite challenging, with a very large state space and practically relevant objectives, underlining the capability of MOGFN-PC to effectively resolve challenging multi-objective scientific discovery tasks.

We are happy to address any further questions or comments from the reviewers to improve the paper and look forward to a fruitful discussion!

---

### Decision · Program_Chairs · 2023-01-20

**Decision:**

Reject

**Justification For Why Not Higher Score:**

- Insufficient novelty
- Evaluations leave out important baselines
- Sensitivity of the method's performance to its input parameters without a clear way to choose these
- Lack of clarity

**Justification For Why Not Lower Score:**

N/A

**Metareview: Summary, Strengths And Weaknesses:**

This paper proposes a new method for sampling points on the Pareto frontier and for performing multi-objective optimization with expensive-to-evaluate objectives. In addition to the standard criteria for these problems, the method aims to generate Pareto-optimal inputs that are different from each other (in the input space), with applications in drug discovery.

The paper is based closely on GFlowNets, recently proposed by Bengio et al. 2021b. This is a method for sampling inputs x approximately proportionally to their reward distribution.

The method in this paper, MOGFN-PC, for generating diverse inputs on the Pareto frontier works as follows:
1. choose a scalarization of the multiple objectives, which depends on a "preference" parameter omega
2. train a GFlowNet to sample inputs x proportionally to this scalarized reward, given the preference parameter
3. To sample candidates, draw the preference parameter at random (the paper proposes using a Dirichlet distribution), then sample from the GFlowNet

The second method in this paper, MOGFN-AL, for doing multi-objective optimization of expensive functions works as follows:
1. Choose a multi-objective BayesOpt acquisition function (the paper proposed expected hypervolume improvement, from Daulton et al., 2022)
2. Use GFlowNets to sample inputs proportional to this acquisition function
3. Evaluate the objective function at these candidates
4. Repeat, supplying these evaluations as input to the BayesOpt method

These methods are designed to work in complex discrete design spaces, such as those arising in molecular discovery.

3 reviewers provided reviews with two reviewers rating the paper a 3 and one rating it an 8. Given this substantial diversity in opinions, I spent additional time reviewing the paper myself.

Strengths
1. Diversity is an important characteristic in drug discovery --- if one candidate compound turns out to be poor as a drug because of some unanticipated constraint, it is useful to have other compounds that are different and thus likely to not also fail this same constraint.
2. The proposed methods outperform baselines in an interesting set of experiments with complex high-dimensional inputs

Weaknesses
1. Lack of novelty. The first proposed method, MOGFN-PC, combines GFlowNets and scalarizations in a straightforward way. The second proposed method, MOGFN-AL, combines GFlowNets and (MOO) Bayesian in a straightforward way. The second method is also close to GFlowNet-AL (Jain et al., 2022).

2. Given how well-studied the space of MOO for expensive functions is, the evaluations leave out many strong baselines. The paper mentions that MORBO (Daulton et al., 2022) and LaMOO (Zhao et al., 2022) are left out because they are designed for continuous spaces, but as the paper describes, there are now-standard methods for embedding molecular spaces into the continuum that could be used (e.g., Gómez-Bombarelli et al. 2018). Given that hypervolume is the quality measure used to measure optimization performance (Fig 2) in this setting, it seems important to include as baselines more BayesOpt methods that explicitly seek to optimize hypervolume.

3. The quality of the results are sensitive to the scalarization chosen and the parameter beta that governs the concentration of the reward density. For the hypervolume, there is a factor of 3 difference between the best and worst out of the 3 values evaluated for beta, and a factor of 4 for the scalarization. Similarly, there is roughly a factor of 2 difference for the reward and R_2 metrics. Moreover, there does not seem to be a principled or robust way to choose these parameters.

4. Lack of clarity. Despite the conceptual simplicity of its approach, the paper is difficult to read.


Gómez-Bombarelli, Rafael, et al. "Automatic chemical design using a data-driven continuous representation of molecules." ACS central science 4.2 (2018): 268-276.